# Phosphite as an engineered niche for *Pseudomonas veronii* in a synthetic soil bacterial community

Clara Bailey,[1] Philip Gwyther,[2] Senka Čaušević,[2] Brandon L. Greene,[1,3] Jan Roelof van der Meer[2]

**ABSTRACT** Bioaugmentation, the process of soil restoration by introducing microorganisms capable of degrading pollutants, is a promising and cost-effective strategy for environmental remediation. Aromatic hydrocarbons, such as benzene, toluene, ethylbenzene, and *p*-xylene (BTEX), are highly toxic environmental contaminants that could be transformed to less harmful products through the inoculation of certain organisms capable of BTEX degradation. However, a barrier to successful bioaugmentation is the inoculant's failure to establish within the resident microbial community. In an effort to improve inoculant proliferation, we have investigated phosphite as a phosphorus source for selective nutrient supply. Phosphite is an inaccessible form of phosphorus to organisms that lack the capacity for phosphite oxidation to phosphate. We introduced a phosphite dehydrogenase-coding gene (*ptxD*) into the genome of the toluene-degrading bacterium *Pseudomonas veronii* 1YdBTEX2 to couple phosphite metabolism and aromatic hydrocarbon clearance. When inoculated in either soil matrix or liquid soil extract, *P. veronii* proliferates in a phosphite- and toluene-dependent manner in both growing and stable synthetic soil microbial communities, although the selective effects of phosphite and toluene were not additive in a carbon-limited context. Once toluene is metabolized, *P. veronii* abundance decays, and the microbial community recovers diversity and abundance resembling the uninoculated controls. Additional members of the microbial community were also enriched in the presence of phosphite, and genomic analysis suggests that these microorganisms utilize an alkaline phosphatase, *phoV*, for phosphite assimilation.

**IMPORTANCE** Bioaugmentation is a promising solution to soil contamination, but its practical application is limited due to poor inoculant establishment in the native soil community. This can often be attributed to low nutrient availability and resource competition with native microorganisms. We proposed the use of phosphite as a selective nutrient source to support the growth of a toluene-degrading bacterium, *Pseudomonas veronii*, in a model soil system. We engineered a strain of this organism that was capable of using phosphite as a phosphorus source and saw that phosphite application enhanced the abundance of the inoculant sixfold within a synthetic soil community. In this study, we present the first investigation of a phosphite selection system in the soil microbiome and characterize the environmental conditions in which it is effective. By demonstrating the potential of formulated nutritional niches in soil microbiome interventions, we provide significant insights into the field of microbiome engineering.

**KEYWORDS** bioremediation, niche engineering, phosphite

Soil contamination poses a considerable threat to human health, ecological health, and food security throughout the world (1, 2). Specifically, aromatic hydrocarbon soil pollutants associated with industrial activity, such as benzene, toluene, ethylbenzene,

**Peer Reviewers** Mary Beth Leigh, University of Alaska Fairbanks, Fairbanks, Alaska, USA; April Armes, Oak Ridge National Laboratory, Oak Ridge, Tennessee, USA

Address correspondence to Clara Bailey, clara.bailey45@gmail.com.

The authors declare no conflict of interest.

See the funding table on p. 14.

and *p*-xylene (BTEX), are acutely toxic and carcinogenic (3–5). Remediation of contaminated soil can be achieved through the process of bioaugmentation, where a microorganism capable of pollutant degradation is inoculated at the affected site. However, this method has variable performance due to poor establishment of the inoculated bacterium into the native soil community. Poor inoculant proliferation can be the result of limited nutrient availability despite having plentiful carbon in the form of the pollutants (6–8), as well as from competition for nutrients and loss of metabolic intermediates (9). Providing a selective carbon substrate and/or selective nutrients might help inoculants to sustain and grow amidst a diverse soil microbiome, at least for the duration of degrading the pollutant by the inoculant.

Previous efforts to enhance bioremediation efficiency have focused on phosphorus (P) supplementation, as P is frequently the limiting nutrient for terrestrial microbes (10). However, inorganic phosphate (Pi), the bioavailable form of P, exhibits poor soil mobility and inefficient uptake, necessitating excessive Pi application, which contributes to eutrophication (11, 12). Furthermore, Pi supplementation in P-replete environments would benefit all native and non-native soil microbes and thus would not provide any specific advantage to the inoculated bioaugmenting microbe. Phosphite (Phi) is an alternative P source that is more mobile in soil but requires enzymatic oxidation by phosphite dehydrogenases (*ptxD*) or alkaline phosphatases (*pho*). We have previously shown that, when applied to soil, the phosphite-oxidizing bacterium *Pseudomonas stutzeri* WM88 (now reclassified as *Stutzerimonas* [13, 14]) is capable of Phi-dependent growth and can oxidize sufficient Phi to support plant growth, which is otherwise not bioavailable and toxic to plants (15, 16). Coupling microbe-selective phosphite oxidation to the utilization of the pollutant as a sole carbon and energy source could, in principle, favor specific proliferation of a bioaugmentation inoculant. While *ptxD* has been studied as a selective marker for transgenic plants in weed-control systems (17, 18) and cyanobacteria as a means of biocontainment (19), there is no current knowledge of whether and how Phi utilization via *ptxD* can function to create a selective nutritional niche for the inoculant in the context of soil microbiome interventions.

The aim of the current work was to test whether Phi-utilization could provide a selective nutritional niche favoring survival and growth of a toluene-degrading bacterium in a defined soil community. We develop a Phi-dependent bioaugmentation strain by introducing the *ptxD* gene from *P. stutzeri* WM88 into the pollutant-degrading bacterium *Pseudomonas veronii* 1YdBTEX2. *P. veronii* 1YdBTEX2 was isolated from jet fuel-contaminated soil (20) and can degrade BTEX as carbon sources. Although the genome of *P. veronii* 1YdBTEX2 contains no genes of any known Phi-oxidizing pathways, it does contain the *phnCDE* operon, known to transport phosphite as well as phosphate and small phosphonates (21, 22). To assess the effect of Phi on the establishment of the *P. veronii ptxD*$^+$ inoculant and toluene degradation rates, we use a model synthetic soil community (SynCom) in soil microcosms as our background. This community consists of 21 soil isolates (Table S1) covering four major bacterial phyla that have reproducible assembly and successional patterns (23). We further contrast two inoculation conditions that result in varying nutritional niche availability: one in which the SynCom simultaneously grows with the *P. veronii* inoculant ("growth," higher nutrient availability) and the other, in which first the SynCom is grown in the microcosms and only then *P. veronii* is inoculated ("stable," lower nutrient availability; Fig. 1). We quantify community and *P. veronii* growth, as well as compositional changes, and measure toluene degradation rates. Our results indicate that Phi enhances *P. veronii ptxD*$^+$ abundance in uncontaminated and toluene-contaminated soil and enhances toluene degradation in liquid soil extract (SE) cultures. Nevertheless, the coupling of Phi oxidation to toluene degradation is not additive for *P. veronii ptxD*$^+$ proliferation in the SynCom-colonized soil microcosms, possibly because of alternative Phi-assimilating pathways being present in the SynCom or saturation of the ecological niche for *P. veronii ptxD*$^+$ within the simplified SynCom.

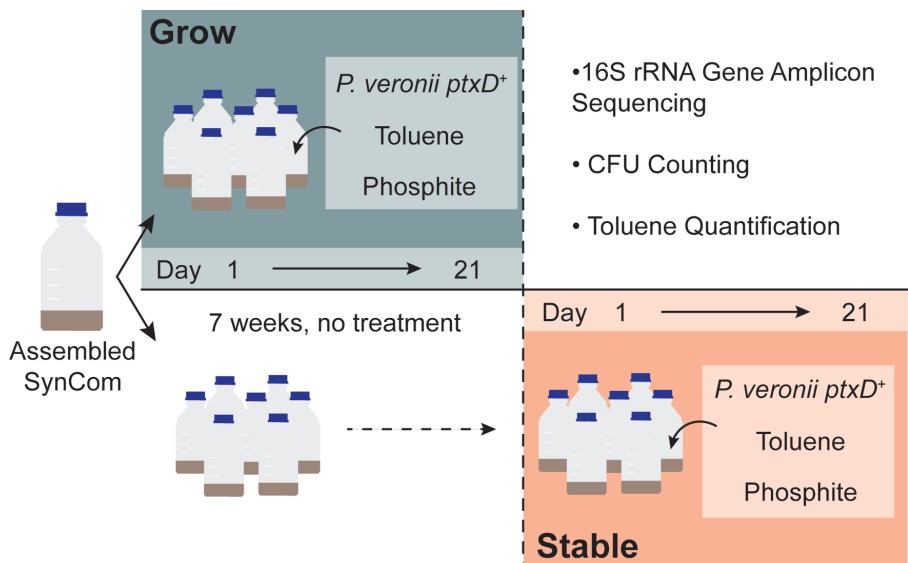

**FIG 1** Overview of soil microcosm experimental layout. Two types of resident communities were produced: growth and stable. The growth phase consists of transferring mature SynCom in a soil microcosm to a fresh soil matrix and SE and adding or withholding all combinations of the inoculant *P. veronii* ptxD⁺, toluene in 2,2′,4,4′,6,8,8′-heptamethylnonane (HMN), and Phi. For the stable phase, the mature SynCom was transferred to fresh soil matrix and SE but was left to develop for 7 weeks without treatment. The previously mentioned conditions were then applied to the stable phase bottles. Timepoints for sampling in each phase are shown on the horizontal axis. On the top right, the sampling methods are outlined.

## RESULTS

### Phi-oxidizing capacity of *P. veronii ptxD*⁺

Recombinant *P. veronii* colonies were verified with colony PCR to confirm the taxonomic identity and demonstrate the insertion of the *ptxD* gene (Fig. S1A). To verify that the *ptxD* insertion conveyed the functional ability to oxidize Phi, *P. veronii* WT and *P. veronii ptxD*⁺ were inoculated Phi- or Pi-containing 21C media. While both the WT and *ptxD*⁺ strains were capable of growth in Pi medium, only *P. veronii ptxD*⁺ displayed growth in the Phi medium (Fig. S1B).

### Phosphite as a selective nutritional niche for inoculant growth in soil microcosms

To evaluate the efficacy of the expressed recombinant *ptxD* gene in *P. veronii* to aid in proliferation in a Phi-amended soil within a microbial consortium background, we inoculated the strain at $10^5$ cells per gram either concomitantly with freshly diluted, assembled SynCom or after full growth of the SynCom in soil microcosms. Assembled SynComs in the absence of *P. veronii,* in the "growth" stage with fresh SE and soil matrix (SM), have sufficient available carbon to allow the community to increase in size and reach maximum cell densities between $10^8$ and $10^9$ CFU/g soil between 24 h and 48 h post-inoculation (Fig. 2A). After 4 days in the growth phase, the community cell viability decreased to $10^7$–$10^8$ CFU/g soil and remained in this range until the beginning of the "stable" phase, 7 weeks after the initial inoculation. During the stable phase, cell densities dropped steadily to $10^6$–$10^5$ CFU/g soil (Fig. 2B), which is consistent with previous observations (23). We also measured the SynCom community composition (in the absence of *P. veronii*) by 16S rRNA gene amplicon sequencing during both growth and stable phases (Fig. 2C and D). After 24 h, the most abundant SynCom member in the growth phase was *Pseudomonas* sp. strain 2, but became less prominent with time throughout growth and stable phases, with a concomitant increase of *Lysobacter*, a facultative predatory bacterium. The ultimate decay of the initial *Pseudomonas* sp.

strain 2 population over the growth and stable phases was correlated with an increase in alpha diversity (Fig. S2). The SynCom dynamics over the growth and stable phases were distinctly clustered by non-metric multidimensional scaling (NMDS) ordination of their relative species abundances (Fig. 2E).

$P.$ $veronii$ $ptxD^+$ inoculated alone in the same microcosm material expands its population size about 50-fold to ca. $10^8$ CFU/g, independent of the presence of Phi (Fig. 2F), indicating that it finds sufficient carbon and nutrient substrates in the soil matrix. In the presence of growing SynCom and in the absence of added Phi, however, $P.$ $veronii$ $ptxD^+$ is not able to expand more than up to ca. $5 \times 10^6$ CFU/g, indicating the competitive effect of the SynCom on $P.$ $veronii$ growth (Fig. 2G). Remarkably, addition of 12 µg Phi per g soil improves $P.$ $veronii$ $ptxD^+$ competitiveness by a factor of 6 ($P = 0.035$ by two-way analysis of variance [ANOVA]), suggesting that Pi was one of the limiting factors for the competitive disadvantage of $P.$ $veronii$ (Fig. 2G). As expected, $P.$ $veronii$ $ptxD^+$ was unable to invade a steady-state grown SynCom in the soil microcosms, since most available carbon has been depleted. But still, addition of Phi enabled the inoculated strain to attain 50- to 100-fold higher abundances than in its absence ($P = 0.048$; Fig. 2H). The application of Phi enabled the inoculant to achieve up to 25% of the total community counts in growth phase, in comparison with only up to 13% in the absence of Phi (Fig. 3A). The inoculant was able to attain up to 0.27% of the total community counts in stable phase in the presence of Phi, and up to 0.16% in the absence of Phi (Fig. 3B). The composition of the SynCom in the presence of Phi was also significantly changed by the presence of $P.$ $veronii$ $ptxD^+$ according to the NMDS ordination ($P = 0.0040$ and $0.0020$ for growth and stable phases, respectively, by two-way ANOVA) in comparison to the absence of Phi in growth phase ($P = 0.185$), but similar to the absence of Phi in stable phase ($P = 0.003$), indicating community response to the competitive advantage imparted to the inoculant by Phi at the growth levels attained from fresh substrate (Fig. S3). These results thus confirmed that Phi can act as a selective nutritional niche to improve inoculant survival amidst a synthetic community that is able to fully colonize the soil.

## SynCom growth and toluene response

Next, we were interested in whether the application of Phi could also improve the rates by which $P.$ $veronii$ would be able to degrade toluene in soil microcosms. To avoid evaporative losses and to decrease toxicity, toluene was introduced into the soil microcosm as a solute in the inert carrier solvent 2,2′,4,4′,6,8,8′-heptamethylnonane (HMN) (24). We then measured toluene metabolism and evaporative losses using headspace solid-phase microextraction (HS-SPME) (25, 26) and gas chromatography-mass spectrometry. Relative abundances of $P.$ $veronii$ $ptxD^+$ were visually higher in the presence of toluene in the microcosms than in its absence (Fig. 3C and D). In the absence of the SynCom, but in the presence of toluene, $P.$ $veronii$ $ptxD^+$ growth was five times higher (Fig. 4A) than in toluene-free microcosms (Fig. 2F), indicating utilization of toluene (Fig. 4B). Toluene loss (Fig. 4B) was significantly greater than the evaporative loss in sterile controls ($P = 0.020$, by two-way ANOVA; Fig. 4C), which retain over 40% of the initial toluene present after 192 h (8 d). The addition of Phi did not affect growth ($P = 0.40$) or toluene ($P = 0.63$) clearance in $P.$ $veronii$ $ptxD^+$mono-colonization of the soil microcosms (Fig. 4A and B). The total SynCom growth without inoculated $P.$ $veronii$ after mixing with fresh soil matrix (growth phase) containing 130 µg toluene per g soil was indistinguishable from growth in soil without toluene or HMN ($P = 0.34$; Fig. 4D), and toluene loads as high as 865 µg/g soil could be tolerated as long as HMN was used as the delivery mechanism (Fig. S4). The community composition in the absence of added $P.$ $veronii$ $ptxD^+$ (Fig. S5) and alpha diversity analysis showed a significant change in richness ($P = 0.026$, Welch's $t$-test), but no change in the Shannon or Simpson diversity indices in comparison to the toluene-free control (Fig. S6A). Toluene exposure during the stable phase had no effect on total cell counts over the course of the 21-day observation period ($P = 0.57$, by two-way ANOVA; Fig. 4E) and a decrease in richness ($P = 4.83 \times 10^{-8}$, Fig. S6A). Based on an NMDS ordination and Bray-Curtis distances, there is no statistically

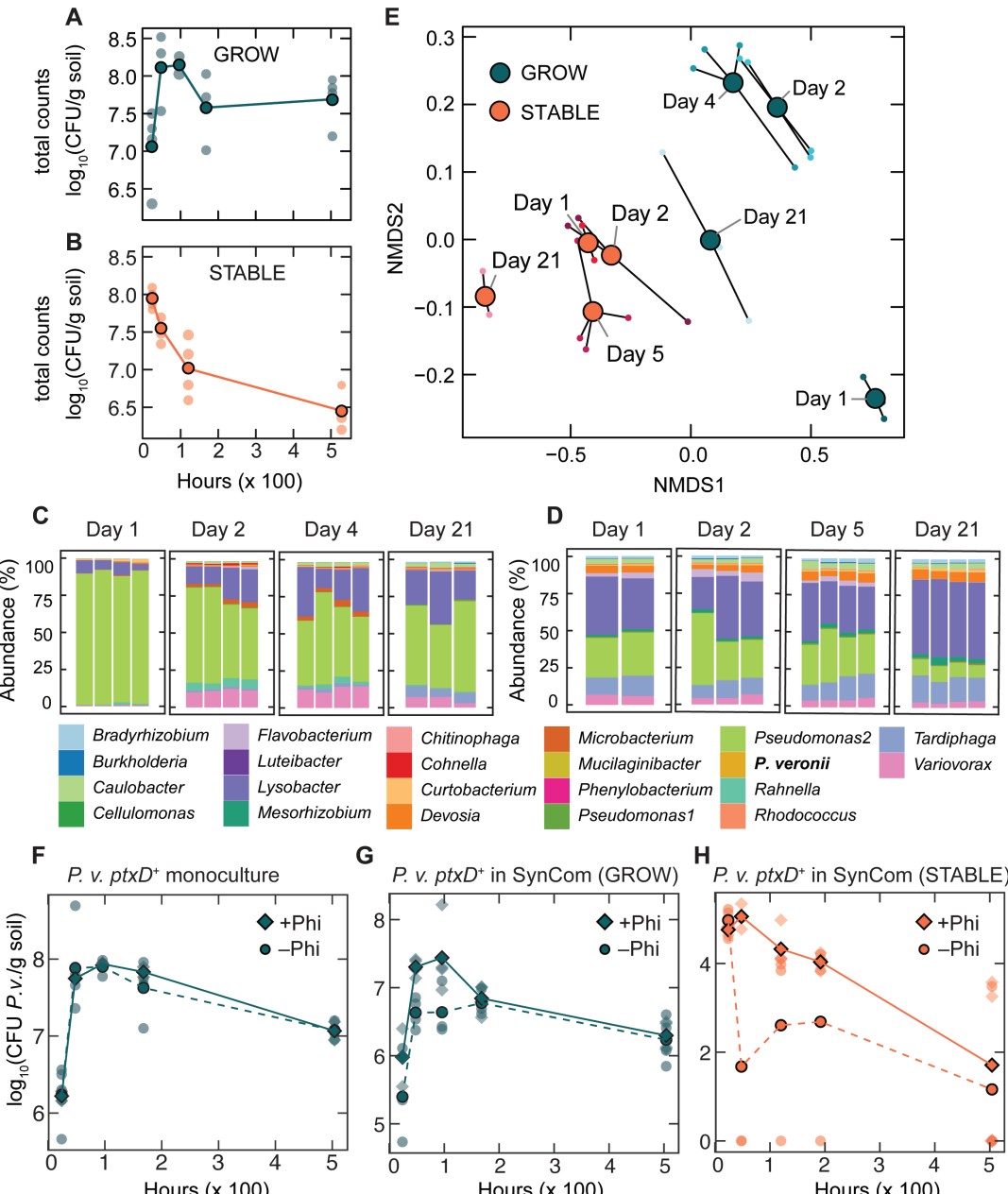

**FIG 2** Phi as a selective niche for *P. veronii* ptxD[+] in soil microcosms. (A) Total CFU per gram of soil over time of the SynCom without *P. veronii* ptxD[+] in the growth phase. (B) Total CFU per gram of soil over time of the SynCom without *P. veronii* ptxD[+] in the stable phase. (C–D) Stacked relative abundances of the SynCom strains in soil microcosm in (C) growth and (D) stable phases. Stacked columns indicate individual biological replicates. SynCom member annotations are indicated below. (E) NMDS analysis of the SynCom species relative abundance data during the growth (dark aqua) and stable (orange) phases of development over time using Bray-Curtis distance values (stress = 0.026). Daily centroids are shown as outlined circles, and replicates are shown as smaller circles, colored according to timepoint and community phase. (F–H) Absolute *P. veronii* ptxD[+] abundance over time in soil microcosm in the presence (dark aqua diamonds) and absence (dark aqua circles) of 12 µg Phi per g soil when inoculated in soil microcosm alone (F), in the presence of growing SynCom (G), and in stable, grown SynCom (H). The average of biological replicates for panels A–B and F–H is shown in dark aqua or orange and individual measurements in translucent aqua or orange.

significant effect of the addition of 130 µg toluene per g soil on the SynCom development, either in growth or stable phases ($P = 0.43$ by permutational analysis of variance [PERMANOVA]; Fig. S7).

In the presence of toluene, the competitive advantage of *P. veronii ptxD[+]* within the soil SynCom was again increased, as expected from toluene being a selective carbon

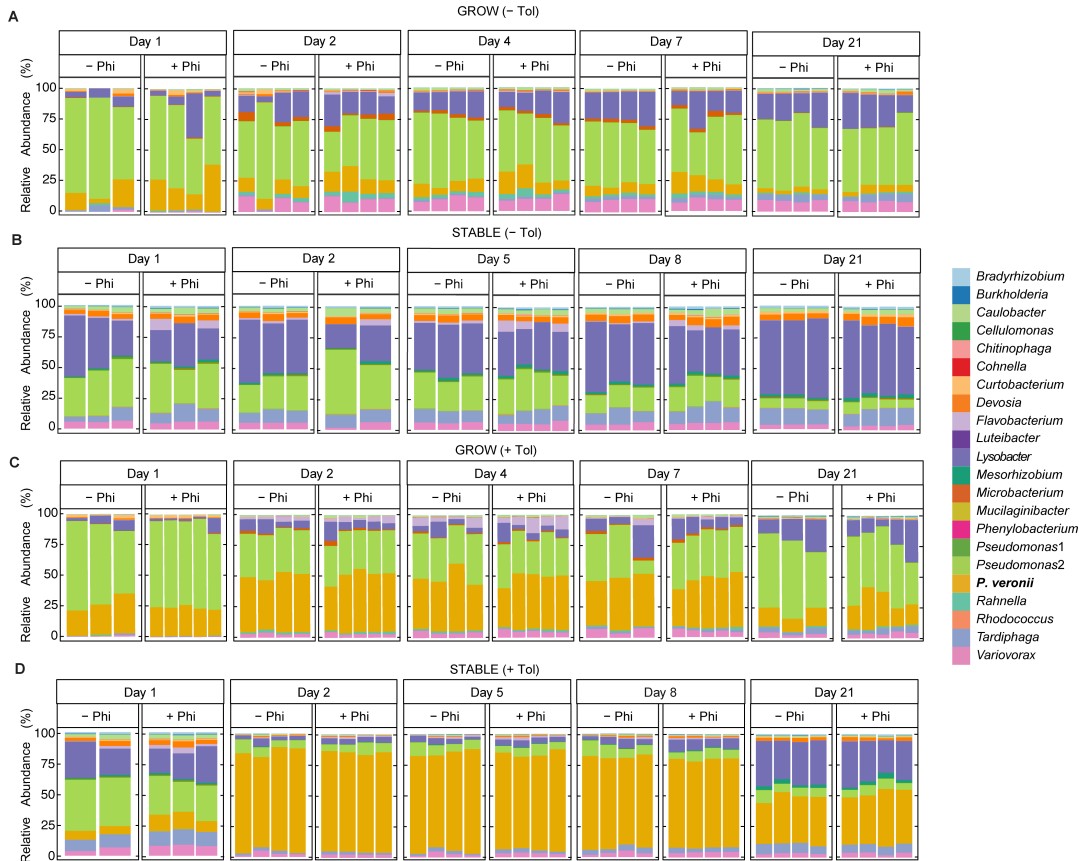

**FIG 3** Stacked relative abundances of the SynCom strains and *P. veronii ptxD* (*P. veronii* in the legend) with and without Phi and toluene. Stacked relative abundances are shown in toluene-free soil microcosm in growth (A) and stable (B) phases, and in toluene-contaminated soil microcosm in growth (C) and stable (D) phases, with and without the application of Phi. Columns indicate individual biological replicates. SynCom member annotations are indicated on the right.

substrate, and population densities were enriched up to 10-fold (in growing SynCom background, day 4) and 1,000-fold (in stable SynCom, day 2; Fig. 3, Fig. 4F and G, $P = 4.1 \times 10^{-4}$ and $2.6 \times 10^{-10}$ by two-way ANOVA for growth and stable phases, respectively). The inoculant contributed up to 46% of the total community counts in the growth phase and up to 81% in the stable phase of SynCom (Fig. 3C and D). Toluene measurements also indicated a slightly faster removal when *P. veronii* was inoculated, allowing the toluene to reach <0.6% of the initial concentration after 48 h (Fig. 4H, Fig. 4I). During the growth phase of SynCom, net toluene loss in the soil microcosm due to the presence of *P. veronii ptxD⁺* at 48 h yields an apparent toluene carbon consumption amount of $5.9 \times 10^{-5}$ g$_C$/g$_{soil}$. This carbon consumption is greater than the carbon associated with *P. veronii* biomass growth between 24 h and 48 h, only $2.3 \times 10^{-8}$ g$_C$/g$_{soil}$, assuming 280 fg C per cell and a carbon to biomass yield of 10%, suggesting that toluene metabolism by *P. veronii* contributes to the growth of other SynCom members, perhaps through cross-feeding, as shown previously (9). Unlike the growth phase, toluene consumption was correlated with *P. veronii* biomass in the stable phase, with a net toluene consumption of $1.2 \times 10^{-5}$ g$_C$/g$_{soil}$ between 24 h and 48 h, a value near the predicted value of $4.8 \times 10^{-5}$ g$_C$/g$_{soil}$ based on *P. veronii* biomass growth between 24 h and 48 h.

## Toluene removal by *P. veronii ptxD⁺* in the presence of phi

In the presence of toluene, applied Phi had no further additive effect on *P. veronii ptxD⁺* proliferation within the SynCom background (Fig. 5 , Fig. 3C and D). Only for the day 21 timepoint for growth phase ($P = 0.019$, Welch's *t*-test) and stable phase ($P = 0.057$, Welch's *t*-test) could a threefold (growth) and twofold (stable) increase in *P. veronii ptxD⁺*

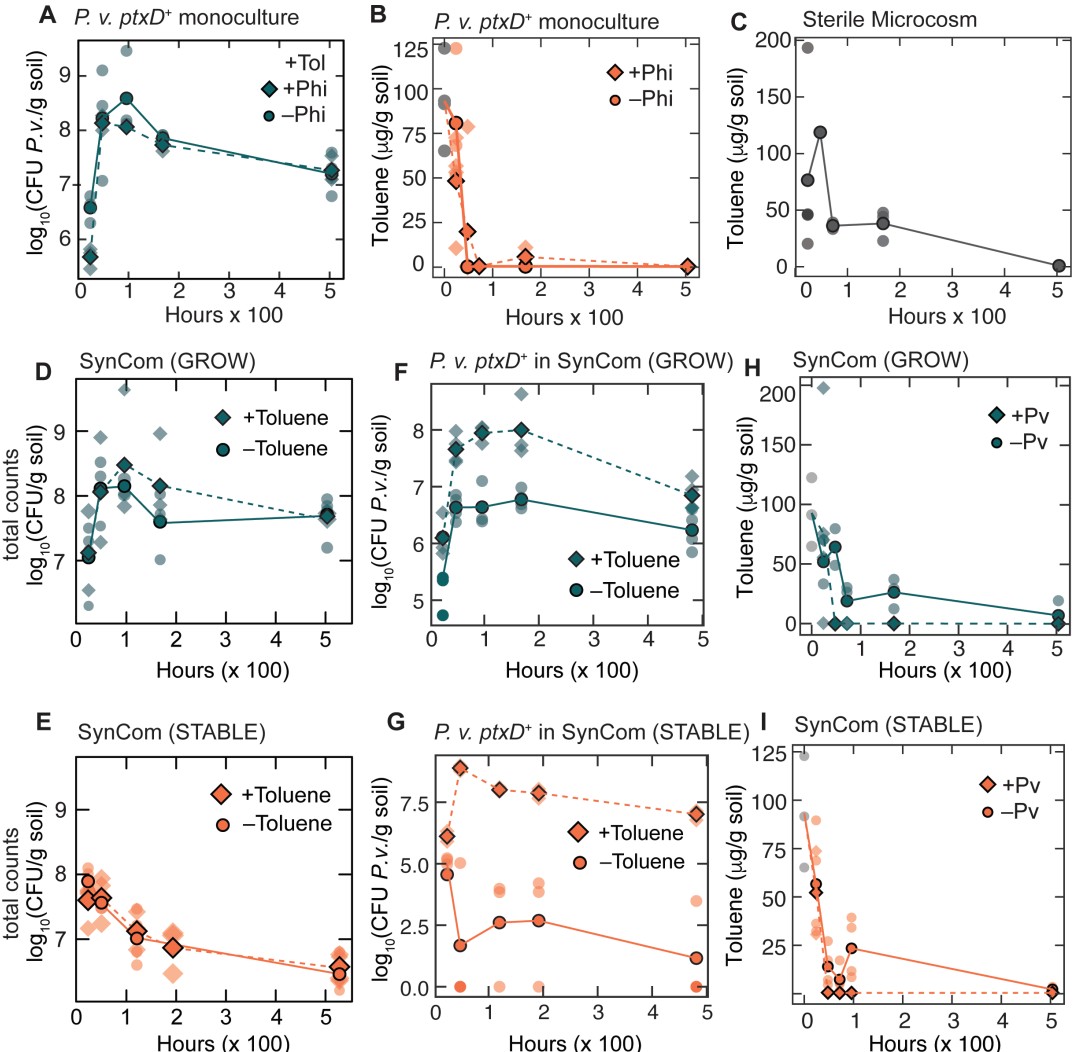

**FIG 4** Effect of toluene exposure on SynCom and *P. veronii* ptxD⁺. (A) Absolute *P. veronii* ptxD⁺ abundance over time in soil microcosm contaminated with 130 µg toluene per g soil in the presence (diamonds) and absence (circles) of 12 µg Phi per g soil. (B) Toluene concentrations over time in soil microcosms inoculated with *P. veronii* ptxD⁺ and 130 µg toluene per g soil in the presence (diamonds) and absence (circles) of 12 µg Phi per g soil. (C) Toluene concentrations over time in soil microcosms without the SynCom or inoculant. (D) Cell densities of SynCom in growth phase in the presence or absence (reproduced from Fig. 2A for comparison) of toluene. (E) Absolute abundances of *P. veronii* ptxD⁺ in toluene-contaminated soil microcosm with (diamonds) and without (circles) toluene in growth phase SynCom ($P = 4.1 \times 10^{-4}$, two-way ANOVA). (F) Toluene concentrations over time in soil microcosms with 130 µg toluene per g soil and growth phase SynCom with (diamonds) and without (circles) *P. veronii* ptxD⁺. (G) Cell densities of SynCom in stable phase in the presence or absence (reproduced from Fig. 2B for comparison) of toluene. (H) Absolute abundances of *P. veronii* ptxD⁺ in toluene-contaminated soil microcosm with (diamonds) and without (circles) toluene in stable phase SynCom ($P = 2.6 \times 10^{-10}$, two-way ANOVA). (I) Toluene concentrations over time in soil microcosms with 130 µg toluene per g soil and stable phase SynCom with (diamonds) and without (circles) *P. veronii* ptxD⁺. Average of biological replicates for panels A–I is shown in dark aqua, orange, or gray and individual measurements in translucent aqua, orange, or gray.

abundance be observed compared to conditions without added Phi. Within the SynCom community, *Pseudomonas* 2 reached the largest absolute population abundance during the growth phase (Fig. S8), suggesting nutritional niche overlap with added *P. veronii* ptxD⁺, whereas *Lysobacter* was the most abundant SynCom member in the late stable phase, potentially feeding off bacterial biomass (Fig. 3C and D). The similarity of the community composition along the growth and stable phase is shown in the NMDS plot (Fig. S9, $P = 0.79$ by PERMANOVA). No Phi effect was observed for toluene clearance, which was complete within the first 48 h in both growth and stable phases with or without Phi (Fig. 5C and D, $P = 0.42$ and $0.16$ for growth and stable phases, respectively,

of timepoints 0–48 h, by two-way ANOVA). The data are consistent with toluene serving a significant role in stimulating *P. veronii ptxD*$^+$ growth, and Pi in the soil microcosm being insufficiently depleted to necessitate Phi metabolism, except when competing among the SynCom for soil microcosm carbon.

To examine how nutrient availability affects toluene clearance in a bioaugmented environment with a lower Pi background, we used liquid SE, which has lower Pi availability and suffers less from evaporative loss of toluene. The SynCom composition was distinct in the SE at all timepoints over the 21-day incubation, relative to the soil matrix microcosm in either growth or stable phases, with a much larger relative abundance of *Rahnella* and *Pseudomonas* strain 1 (Fig. S10). We also increased the toluene concentration from 130 µg toluene per g soil to 600 µg toluene per mL to ensure abundant substrate and enhance a potential advantage for Phi on *P. veronii* growth (assuming Pi-limitation would show more rapidly at higher net *P. veronii* growth). Indeed, *P. veronii ptxD*$^+$ exhibited higher absolute abundances in toluene-contaminated SE medium in the presence of Phi for the first 48 h after inoculation than in its absence (Fig. 6A and B, $P = 9.1 \times 10^{-5}$, two-way ANOVA). This threefold increase of *P. veronii ptxD*$^+$ abundance at 24 h coincided with nearly threefold lower toluene concentrations in the Phi-amended SE cultures at the 24 h timepoint (Fig. 6C, $P = 0.050$, Kruskal-Wallis test). Using an exponential fit of the data, toluene consumption was $0.06 \pm 0.03$ h$^{-1}$ in the presence of Phi and $0.04 \pm 0.01$ h$^{-1}$ in the absence of Phi. Toluene-exposed and inoculant-induced SynCom cultures demonstrated a drop in cell viability at the 24 h timepoint that was completely recovered in the cultures supplemented with Phi (Fig. S11; $P = 0.018$, Kruskal-Wallis test).

## Phi oxidation by SynCom members

Despite improved establishment of *P. veronii ptxD*$^+$ in Phi-supplemented soil microcosm and faster toluene clearance, alternative Phi-metabolism pathways in SynCom members could compromise the Phi-dependent bioaugment stimulation. We examined the SynCom members' annotated genomes for homologs to known Phi-oxidizing pathways, encoded in the *ptx*, *htx*, and *phn* operons (27, 28), using the NCBI blastp algorithmm (*E* value cut-off of $1 \times 10^{-50}$; Table S3). Sequence alignments of PtxD with glyoxylate/hydroxypyruvate reductase B found in *Burkholderia* and *Variovorax* show similar percent sequence identities as other NAD$^+$-dependent D-hydroxy acid dehydrogenases of known function have with PtxD (29). Another pathway for Phi-oxidation in *Escherichia coli* is the bacterial alkaline phosphatase (APase) *phoA*, which catalyzes Phi disproportionation to phosphate and molecular hydrogen (30). While APases primarily act on phosphomonoesters, their substrate promiscuity is well established and thought to play an important role in the biogeochemical P cycle (31–33). We compared SynCom relative abundances in soil microcosms with or without addition of Phi (Fig. 7A) and in the presence of *P. veronii ptxD*$^+$ (Fig. 7B) to identify any enriched SynCom members in soil microcosms. Community analysis showed elevated relative abundance of *Chitinophaga* and *Luteibacter* in SynCom supplemented with Phi in stable phase (Fig. 7A). In the presence of *P. veronii*, *Flavobacterium* was enriched in the soil microcosm (Fig. 7B). Of the APases identified in the annotated genomes of the SynCom members, *phoV* was the only one shared by *Flavobacterium*, *Chitinophaga*, and *Luteibacter* (Fig. 7C). Although *phoV* was also identified in the *Mucilaginibacter* genome, this member did not reach abundances above the detection limit for any soil microcosm condition. The protein encoded by the *phoV* gene has been investigated in the cyanobacterium *Synechococcus elongatus* and was shown to be a Pi-irrepressible alkaline phosphatase, the same enzyme family of the widely distributed and Pi-independent phosphatase *pafA* (32, 34, 35). Whether *phoV* is not only present in the genome of these organisms but also expressed and active as PhoV is an area of future study.

## DISCUSSION

In this study, we have assessed the effectiveness of Phi as a nutrient for selectively promoting the growth of *P. veronii ptxD*$^+$, a genetically engineered Phi-oxidizing strain of

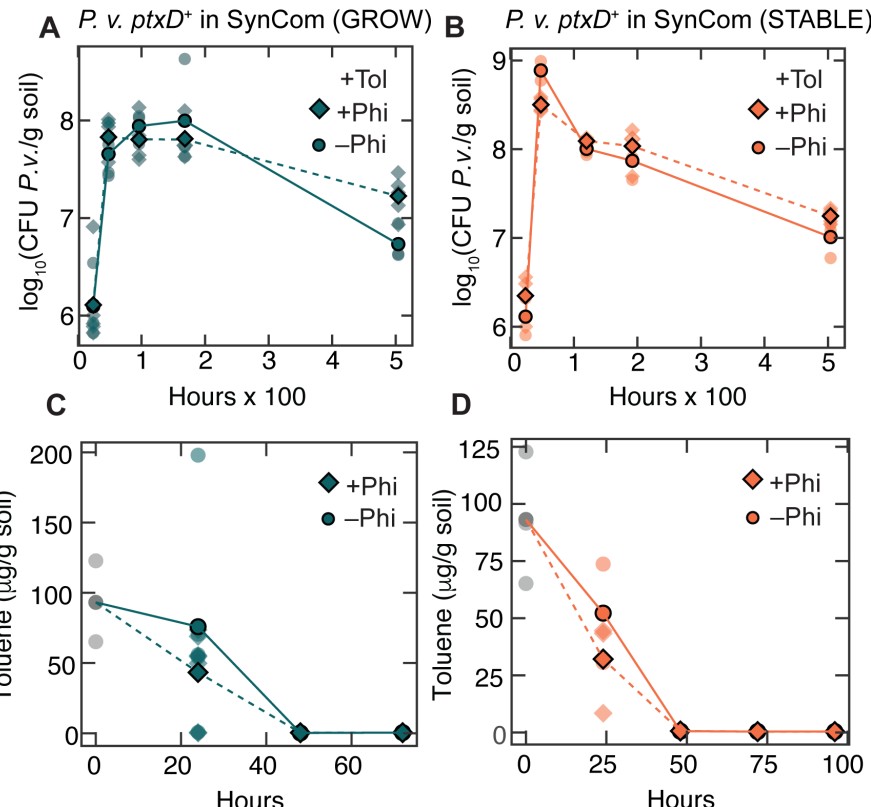

**FIG 5** Phi effect on toluene-remediation system. Absolute abundances of *P. veronii* ptxD$^+$ in toluene-contaminated soil microcosm with and without 12 µg Phi per g soil in the growth (A) and stable (B) phases of SynCom. Timepoints of increased *P. veronii* ptxD$^+$ abundance in toluene-contaminated microcosm were found at day 21 of growth phase ($P = 0.019$, Welch's $t$-test) and stable phase ($P = 0.057$, Welch's $t$-test). Toluene concentrations in soil microcosm for (C) growth ($P = 0.42$ for 0–48 h, by two-way ANOVA) and (D) stable phases ($P = 0.16$ for 0–48 h, by two-way ANOVA) SynCom with *P. veronii* ptxD$^+$ with (diamonds) and without (circles) Phi. The averages of biological replicates for panels A–D are shown in dark aqua or dark orange, with individual measurements in translucent aqua or orange.

the toluene-degrading soil bacterium *P. veronii* 1YdBTEX2. Our results demonstrate that the presence of Phi leads to increased abundance of the inoculant, with the extent of enhancement depending on the growth stage of the resident microbial community, and indicates that competition for Pi is one reason for poor colonization of *P. veronii* amidst growing soil SynCom. As expected from the soil SynCom composition, toluene itself also acts as a selective nutritional niche (in the form of a carbon substrate) for better growth of *P. veronii* in the background of SynCom. The combination of both toluene and Phi did not further improve the growth of the *P. veronii* ptxD$^+$ inoculant, although it did result in slightly faster toluene metabolism.

The larger selective effect of toluene in comparison to Phi suggests that the soil microcosm model used in this study is C-limited, rather than P-limited, which may not be representative of potential contaminated sites (36, 37). This reduced response to Phi in combination with toluene addition is particularly evident in the early growth phase, when Pi availability is highest upon transfer to fresh substrate, and no Phi selection is observed until day 21. To show that Phi addition can further promote the establishment of the inoculant in the presence of toluene, we used SE liquid experiments, where low Pi concentrations could be maintained and Phi was important to overcome P-limitations. Although Pi does not act as an enzymatic inhibitor to PtxD (38), the competitive advantage of Phi in the presence of high Pi availability will be strongly dependent on the simultaneous availability of carbon substrates and competitive effects by the background community. Under conditions of highly available and selective carbon substrate

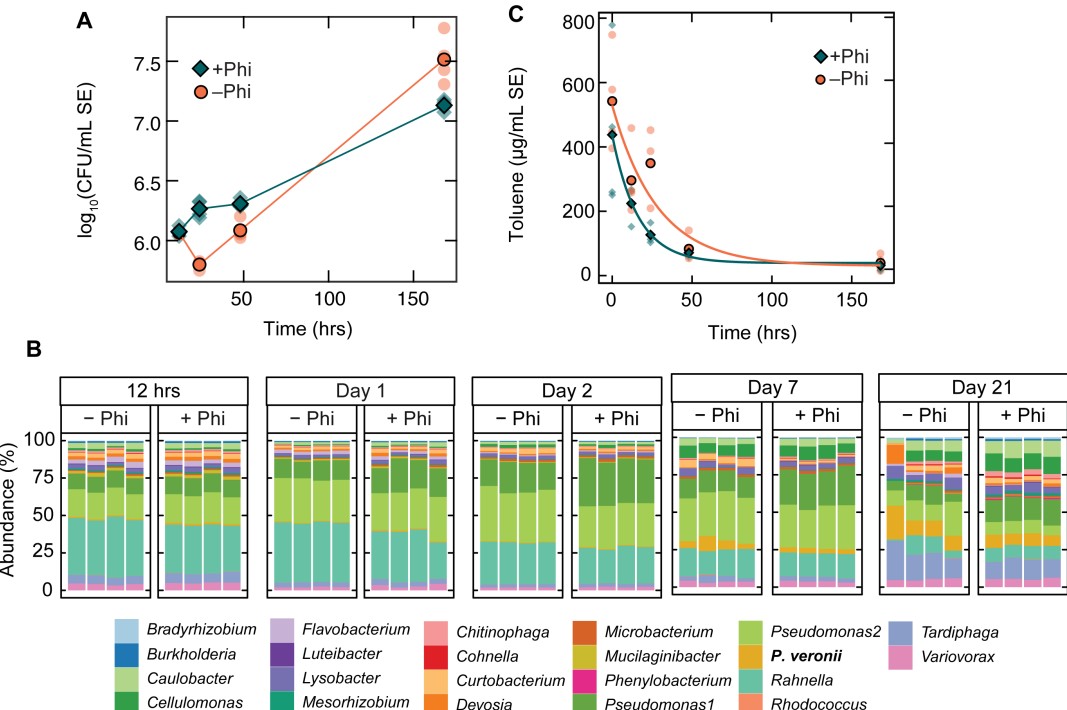

**FIG 6** The Phi effect in liquid SE. (A) Absolute abundances of *P. veronii* ptxD⁺ in toluene-contaminated, SynCom-containing SE with (dark aqua diamonds) and without Phi (orange circles; $P = 9.1 \times 10^{-5}$, two-way ANOVA). Outlined points represent the average value of biological replicates. (B) Stacked relative abundance of SynCom members over time in SE medium in the presence of *P. veronii ptxD*⁺ with and without Phi. Columns indicate individual biological replicates. SynCom member annotations are indicated below. (C) Toluene concentrations with (dark aqua diamonds) and without (orange circles) Phi measured in toluene-contaminated SE containing SynCom and *P. veronii* ptxD⁺. Outlined points represent the average value of biological replicates. An exponential fit of the data is shown as solid lines, with kinetic constants of $0.04 \pm 0.01$ h⁻¹ in the absence of Phi and $0.06 \pm 0.03$ h⁻¹ in the presence of Phi.

(such as toluene addition here and no toluene metabolizers in the SynCom community), the toluene-degrading inoculant has a sufficient competitive edge to outgrow and uptake available Pi from the habitat even in the presence of growing resident bacteria. Toluene degradation in *P. veronii* 1YdBTEX2 is mediated by the products of the *ibp*, *dmp*, and *ttg* genes, which are all located on the chromosome-2-replicon and strongly induced in the presence of BTEX (39). In the absence of a selective carbon substrate, the inoculant improves competitive growth on background carbon by assimilating Phi. Therefore, the efficacy of a Phi-directed remediation system is promising but contingent on the environmental context. Although for simplicity of the study, we focused entirely on simulating the effects of aerobic soil conditions, real contaminated environments pose the additional complexity of alternating oxic/anoxic conditions and/or water-saturated zones. It is interesting that *P. veronii* 1YdBTEX2 is capable of growth under denitrifying conditions and could thus potentially survive or proliferate during periods of anoxic conditions, provided that nitrite or nitrate is available (40). Anoxic conditions themselves, however, are incompatible with BTEX degradation by *P. veronii* 1YdBTEX2, since the degradation pathway is mediated by dioxygenase enzymes (IbpAaAbAcAd). This system might thus be most appropriately applied to aerobic soil sites with poor pollutant availability and low bioavailable P, an increasing proportion of the global terrestrial biosphere (41, 42).

This study provides additional insight into the establishment of an inoculant into a soil microbiome through the quantification of toluene concentrations with HS-SPME. Notably, the observed net toluene consumption in growth phase does not correspond to equivalent growth in *P. veronii ptxD*⁺ biomass. This could be due to metabolite leakage commonly found in microbial communities as part of the mutually beneficial process of cross-feeding (43, 44). Cross-feeding of toluene metabolites has been shown previously

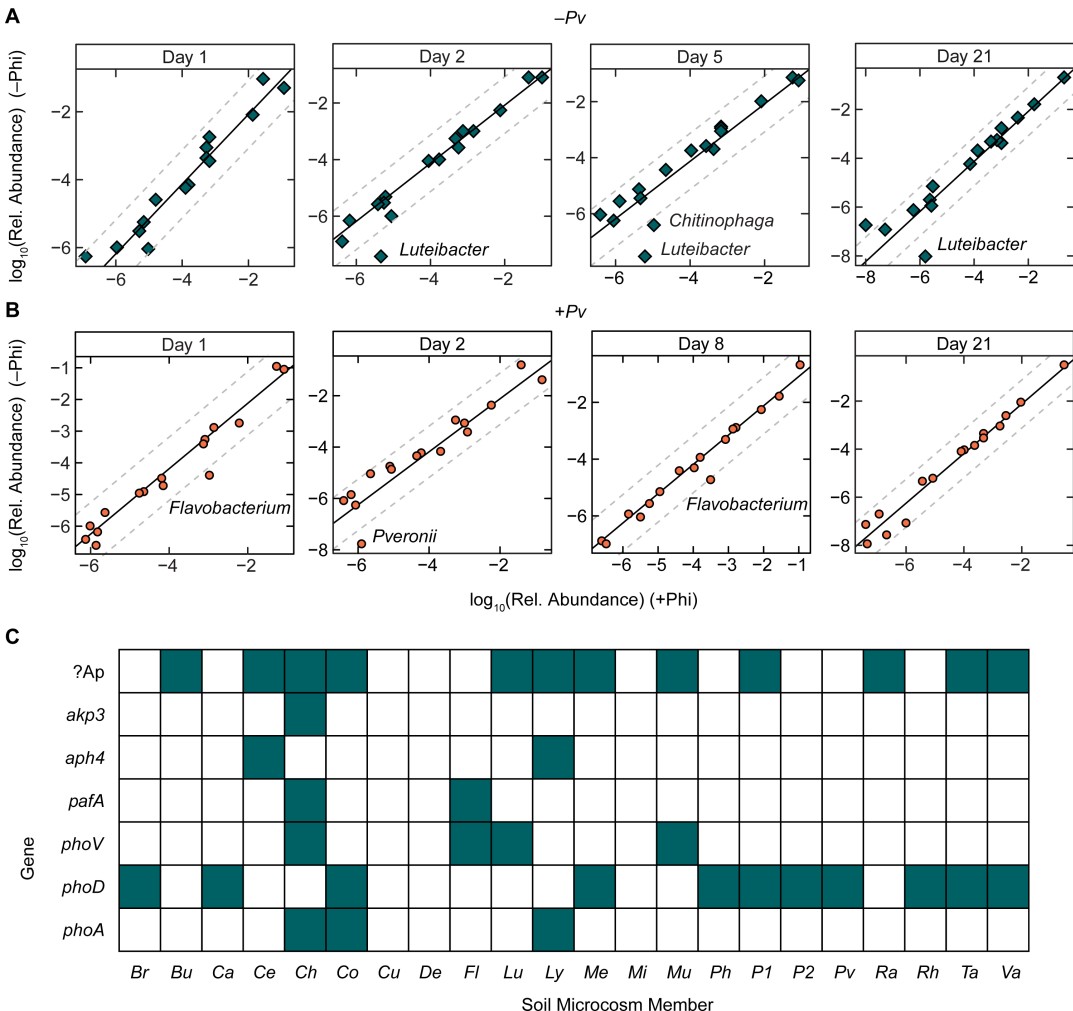

**FIG 7** The Phi effect on SynCom development and alkaline phosphatase gene presence in SynCom members. (A) Linear regression model comparing strain relative abundance data for SynCom in soil microcosm with (x-ordinate) vs without (y-ordinate) Phi application. Enriched strains, those that deviate more than 1 log away (dashed lines) from the linear model (solid line), are highlighted by their strain name. (B) Linear regression model comparing strain abundance data for SynCom and *P. veronii* ptxD$^+$ in soil microcosm with or without Phi application. (C) Alkaline phosphatase genes present in the SynCom member annotated genomes, as well as *P. veronii* ptxD$^+$. The SynCom members (and *P. veronii* ptxD$^+$) are all represented by the first two letters of their genus, except for *Pseudomonas* sp.

in toluene-contaminated, naturally derived soil communities inoculated with *P. veronii* 1YdBTEX2 (9). A similar effect may be present in relation to Phi oxidation by *P. veronii* ptxD$^+$, through the transformation and "leakage" of Phi to broadly available Pi, facilitating growth of other SynCom members and aggravating the resource competition from the perspective of the inoculant.

The utility of Phi-selective stimulation of inoculants in soil may be compromised to some extent by the potential for (alternative) Phi oxidation pathways amidst soil community members. Growth of two SynCom members in the presence of Phi suggests that Phi assimilation is more widely distributed in soil communities than the 1.5% of sequenced bacteria and archaea predicted to assimilate Phi based on the prevalence of *ptxD* (45). Given the increasing interest in the use of Phi as a selective fertilizer for transgenic plants, the Phi-oxidizing capabilities of native soil communities should be considered, especially in Pi-abundant environments (12, 17, 18). Phi-oxidizing alkaline phosphatases have only recently been proposed as an alternative to *ptxD* for Phi selection in transgenic plants (46). The performance of Phi-oxidizing alkaline phospha-tase selection systems in soil communities, in comparison with *ptxD*-based selection

systems, and the impact of hydrogen generation on neighboring microbial metabolism merit further study.

## MATERIALS AND METHODS

Construction of the *P. veronii ptxD*[+] mutant is detailed in the Supplemental Information.

### Soil microcosm preparation and inoculation

Soil microcosms were prepared as previously described with minor modifications noted below (23). SM was made from riverbank sediment by the Sorge river in Lausanne, Switzerland (46°31′22.4″N 6°34′31.7″E). The sediment was dried and double sieved to obtain 0.5–3 mm-sized particulates. The SM was autoclaved, and a week later was divided into 85 g portions in 500 mL Schott glass bottles (90 g for the initial SynCom inoculation). The individual flasks were closed and autoclaved twice to ensure all spores were killed, and a bottle was tested for sterility by adding 10 g of SM to 10 mL sterile 0.2% (wt/vol) tetrasodium-pyrophosphate decahydrate (pH 7.5, Sigma Aldrich), vortexing for 1 min, and spreading 100 µL of the supernatant onto a Reasoner's 2A (R2A) agar plate. SE was collected from the Dorigny Forest in Lausanne, Switzerland (46°31′16.4″N 6°34′43.0″E). Topsoil was collected and mixed in a 1:1 ratio with tap water. The mixture was autoclaved and left overnight, then decanted into centrifuge tubes and centrifuged at $5,000 \times g$ for 15 min. The supernatant was combined, mixed, and pooled into 1 L Schott bottles before autoclaving and testing for sterility by plating 100 µL onto an R2A agar plate.

The SynCom was assembled from plates (Fig. S12) of the 21 isolates individually as described previously (23). The soil microcosms were initiated by inoculating 90 g of SM with 10 mL of SE containing $10^7$ cells mL$^{-1}$ (total) of the SynCom community, with each member in equal proportion from pure starter cultures, as measured by OD$_{600}$. After inoculation, the bottles were mixed on a horizontal roller mixer for 20 min at 80 rpm and incubated at room temperature in the dark for 2 months with periodic rotation. A sufficient quantity of this mature SynCom soil microcosm was prepared to yield a combined 400 g and was mixed with a sterile beaker and stirrer prior to use in inoculating experimental microcosms. To inoculate an individual microcosm, 6 g of the combined mature SynCom was added to 85 g of sterile SM, along with 10 mL of sterile SE. Soil microcosms were either used for inoculation experiments directly ("growth," Fig. 1) or were mixed on a horizontal roller mixer (20 min at 80 rpm) and incubated at room temperature in the dark for 7 weeks with periodic rotation before being inoculated with *P. veronii* ("stable," different permutation conditions see below; Fig. 1).

For experiments done exclusively in SE, $10^7$ cells mL$^{-1}$ of the same equal abundance SynCom community used for soil matrix was added from a 100 µL suspension in 0.9% NaCl to 10 mL of sterile SE (adjusted to pH 8.6) in a 50 mL Falcon tube and vortexed to mix. Falcon tubes were then placed in a room temperature incubator at 100 rpm and incubated for 2 weeks before a 1:10 dilution of mature SynCom into fresh sterile SE.

### Soil microcosm conditions, inoculations, and sampling

The soil microcosms were prepared for two experiment phases: "growth" and "stable" (Fig. 1). The growing phase exposed SynCom to all permutations of the following treatments: the inoculation of *P. veronii ptxD*[+], the introduction of toluene, and the addition of Phi, directly after dilution of the mature SynCom of the first phase into fresh SM and SE. Stable phase experiments similarly involved transferring the SynCom to fresh SM and SE but instead included incubating these bottles in the dark at room temperature for 7 weeks (as outlined above) before exposing the SynCom microcosms to the treatment conditions and *P. veronii ptxD*[+] inoculation. For each treatment, four replicate microcosms were prepared.

To inoculate *P. veronii ptxD*$^+$, the mutant was grown on M9 plates with 20 µg/mL gentamicin and toluene as the carbon source in a toluene chamber. Individual colonies were then picked, inoculated in 10 mL of liquid M9 medium with 20 µg/mL gentamicin and 15 mM succinate, and grown overnight at 30°C while shaking at 180 rpm. Inoculants were thus not pre-adapted to toluene before their addition to the soil microcosms. Prior to inoculation in the soil microcosms, the entire culture was spun down at 4,025 × *g* for 4 min and washed 3× with sterile 0.9% NaCl. The cell density was measured using absorbance at 620 nm, and the culture was appropriately diluted to allow for the addition of 10$^5$ cells of *P. veronii ptxD*$^+$ g$^{-1}$ SM in a total volume of 100 µL.

Toluene (Honeywell), used for simulating contamination in soil matrix, was prepared in a 7.5% (v/v) solution with HMN (Sigma Aldrich), and the SE was amended with 2% (v/v) of the toluene solution and vortexed briefly before addition to the SM (i.e., 0.2 mL in 10 mL SE total). Phi was added directly to the SE to a final concentration of 1.3 mM from a 1.3 M Phi stock (pH 5, Sigma Aldrich) and was filter sterilized prior to soil matrix inoculum preparation. Once the bottles were inoculated with SynCom and all perturbations of the soil experimental conditions were applied, the bottles were rolled to mix for 20 min. Control microcosms without SynCom, but only *P. veronii ptxD*$^+$ or sterile bottles, were subjected to the same conditions. For the SE liquid culture experiments, toluene was introduced in a 35% (v/v) solution with HMN, and Phi was introduced directly to the SE. After applying these conditions, SE microcosms were vortexed for 30 s.

Timepoints for the growing phase were taken 24 h, 48 h, 96 h (4 d), 168 h (7 d), and 504 h (3 weeks) after inoculation. Timepoints for the stable phase were taken 24 h, 48 h, 120 h (5 d), 192 h (8 d, except for the SynCom-only control), and 504 h (3 weeks). Timepoint measurements consisted of CFU counting, the collection of cells for later DNA extraction and 16S rRNA gene amplicon sequencing, and the collection of soil microcosm samples for toluene quantification. First, a sterile inoculation loop was used to collect 100 ± 10 mg of SM + SE in a 1 mL crimp vial and was immediately stored at –80°C for toluene analysis. Then, using a Sterileware spatula (SP Bel-Art), 10 g of microcosm material was removed from the bottle and placed in a 50 mL sterile Falcon tube with 10 mL of 0.2% pyrophosphate solution. The tube was vortexed at maximum speed for 2 min to separate cells from the soil matrix, then the particles were allowed to settle for 1 min. The top layer (~5 mL) of the suspension was removed and placed in a 15 mL sterile Falcon tube. From this, a 100 µL aliquot was removed, subjected to several 1:10 serial dilutions, and plated on R2A agar plates for CFU counting. The remaining cell suspension was spun down at 4,025 × *g* for 4 min, then the supernatant was removed, and the cells were resuspended in ~0.5–1 mL of the pyrophosphate solution and placed in a new 1.5 mL Eppendorf tube. The cells were spun down once more, and the supernatant was removed. Tubes containing the cell pellet were frozen in liquid nitrogen and stored at –80°C for subsequent DNA isolation (see below).

For the SE liquid culture experiments, CFU counts, community composition, and toluene were also measured. Prior to sampling, the culture vials were vortexed at maximum speed for 30 s. Toluene samples were taken as 10 µL SE culture and were placed in crimp vials and stored at –80°C. A 100 µL aliquot was taken directly from the culture for CFU counting as described above. Finally, 1 mL of culture was spun down at 6,000 × *g* for 4 min, the supernatant was removed, and the cell pellet was snap frozen with liquid nitrogen and stored at –80°C until DNA isolation.

The methods for SynCom community analysis and identification of *P. veronii* from 16S rRNA gene sequencing data are detailed in the Supplemental Information. In short, since the genomes of SynCom members and *P. veronii* are available, we directly search for strain-specific signatures in the amplified 16S rRNA gene fragments and correct for the number of 16S rRNA gene copies per strain to report relative abundances. Since all analyses are based on cells separated from the soil matrix and all SynCom members are culturable, we can transform to absolute population abundances by multiplying the (genome-copy number corrected) relative abundances from 16S rRNA gene amplicon sequencing with the total viable counts in the same samples (CFU per

mL). Toluene quantification by gas chromatography-mass spectrometry (GC-MS) and statistical analysis is detailed in the Supplemental Information.

## ACKNOWLEDGMENTS

This work was supported by the Swiss National Center for Competence Research *NCCR Microbiomes* (grant number 180575 to J.R.v.d.M.). C.B. gratefully acknowledges financial support for this work from the Fulbright U.S. Student Program, which is sponsored by the U.S. Department of State and the Swiss Federal Commission for Scholarships (FCS). Its contents are solely the responsibility of the author and do not necessarily represent the official views of the Fulbright Program, the Government of the United States, or the FCS. B.L.G. acknowledges a UCSB Faculty Research Grant and the Reagents Junior Faculty Fellowship for financial support.

The authors acknowledge the support of the Lausanne Genomics Facility for the 16S rRNA gene amplicon sequencing. C.B. gratefully acknowledges Andrew Quinn for their invaluable knowledge and support in the operation and interpretation of the GC-MS data. C.B. also gratefully acknowledges Bouke Bentvelsen for supplying a photo of *Phenylobacterium*. We gratefully acknowledge Prof. Wifred van der Donk for providing the pET-15b 17X-PTDH plasmid.

## AUTHOR AFFILIATIONS

[1]Department of Chemistry and Biochemistry, University of California, Santa Barbara, California, USA

[2]Department of Fundamental Microbiology, University of Lausanne, Lausanne, Switzerland

[3]Biomolecular Science and Engineering Program, University of California, Santa Barbara, Santa Barbara, California, USA

## AUTHOR ORCIDs

Clara Bailey http://orcid.org/0000-0003-4274-5070
Philip Gwyther https://orcid.org/0009-0001-7258-1989
Senka Čaušević http://orcid.org/0000-0001-7930-3968
Jan Roelof van der Meer https://orcid.org/0000-0003-1485-3082

## FUNDING

| Funder | Grant(s) | Author(s) |
| --- | --- | --- |
| Fulbright U.S. Student Program | | Clara Bailey |
| NCCR Microbiomes | 180575 | Jan Roelof van der Meer |

## DATA AVAILABILITY

The raw reads of the 16S rRNA gene V3–V4 amplicon sequencing for the communities in this study are available from the Short Read Archive under BioProject Number PRJNA1136878 for soil microcosms experiments (SM + SE) and BioProject Number PRJNA1134716 for liquid SE experiments. Cleaned strain abundance data and all other numerical data underlying the Figures, as well as any scripts used for data analysis and Figure generation, are available for download from Zenodo (47).

## ADDITIONAL FILES

The following material is available online.

## Supplemental Material

**Supplemental Information (mSystems00061-25-S0001.docx).** Tables S1–S3, Figures S1–S12, and supplemental methods.

## Open Peer Review

**PEER REVIEW HISTORY (review-history.pdf).** An accounting of the reviewer comments and feedback.

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
