## [Reviewer comments · mSystems]

Phosphite as an Engineered Niche for *Pseudomonas veronii* in a Synthetic Soil Bacterial Community

Clara Bailey, Philip Gwyther, Senka Causevic, Brandon Greene, and Jan Roelof van der Meer

Corresponding Author(s): Clara Bailey, University of California Santa Barbara

Review Timeline:

Submission Date:	January 14, 2025
Editorial Decision:	March 9, 2025
Revision Received:	May 1, 2025
Editorial Decision:	May 21, 2025
Revision Received:	July 2, 2025
Accepted:	July 17, 2025

Editor: Hans Bernstein

Reviewer(s): Disclosure of reviewer identity is with reference to reviewer comments included in decision letter(s). The following individuals involved in review of your submission have agreed to reveal their identity: Mary Beth Leigh (Reviewer #1); April Armes (Reviewer #4)

Transaction Report:

DOI: <https://doi.org/10.1128/msystems.00061-25>

Re: mSystems00061-25 (Phosphite as an Engineered Niche for *Pseudomonas veronii* in a Synthetic Soil Bacterial Community)

Dear Dr. Clara Bailey:

Thank you for the privilege of reviewing your work. First of all, I must apologize for the time that took to provide you with information on the status of your research manuscript. Referees are excellent researchers, but also very busy individuals. The four referees who examined your work found the topic of interest, but also made recommendations to improve the clarity of the document. I now ask you to consider their recommendations when revising this version of the manuscript. Below you will find instructions from the mSystems editorial office, and the reviewer comments.

Revision Guidelines

Sincerely,
Jorge Rodrigues
Editor
mSystems

Reviewer #1 (Comments for the Author):

In this manuscript, the authors demonstrate a clever strategy for increasing the success of bioaugmentation for toluene biodegradation. The method involves introduction of a rare phosphite dehydrogenase-coding gene into a toluene-degrading *Pseudomonas* strain, which enables it to overcome P limitation by accessing added phosphite, which other organisms are

generally less able to access. The authors demonstrated success of this strategy by performing a series of experiments on this using an artificial microbial consortium. The discussion rightly addresses the potential limitations of this method in situ, since not all sites will have appropriate conditions. Nonetheless, this is an exciting and valuable contribution to the bioaugmentation literature.

The study appears technically sound and well considered, with ample evidence provided to confirm the effects. Statistical analyses are used effectively in many places, but there remain a few areas where some additional statistical support is needed to support some statements (noted below).

Overall, this is a strong manuscript, in my opinion. A few minor suggestions are listed below for improvements in clarity of wording and statistical support.

L109-110 "organism species" is awkward, I suggest revising to "taxonomic identity" here

Suppl Fig 9 - the naming convention in the figure legend is inconsistent for *P. veronii* - the other organisms are all listed as genus names. Consider making it consistent or otherwise using correct notation (*P. veronii*).

It would also be helpful to highlight the study organism in the legend perhaps in bold or with an "*" to help readers quickly locate it.

Check and do the same for other similar figures.

L149-150 - can you provide statistics to show if the 50 to 100-fold change is significant or not? It might not be. The stats provided on community shifts in the following sentences are helpful. Nonetheless, you should provide some stats here for the 50-100 percent statement, even if not significant.

L161 - modify to "...we were interested in whether..."

L164 - were any controls run to examine the effects of 2,2',4,4',6,8,8'-heptamethylnonane (HMN) alone?

L166-168 - this sentence is confusing, especially what the "its" means in "in its absence". Clarify if you're referring to the absence of toluene, *P. veronii*, the SynCom, etc.

L167-174 - In this passage, there are many statements here about differences in growth, toluene clearance rate, etc. but there are no stats presented. The figures (Fig. 3) are compelling, but statistical analyses are also needed in order to claim that these trends are truly different.

Also, be cautious about the term "rate" since that's a specific measurement that is expressed in units of toluene over time, vs. just a trend of levels over time. You may want to just discuss toluene loss (rather than rate) if you did not calculate rates here.

L278 - "contamination sites" should be "contaminated sites"

Reviewer #2 (Comments for the Author):

Review for "Phosphite as an Engineered Niche for *Pseudomonas veronii* in a Synthetic Soil Bacterial Community"

This study tested whether phosphite (Phi) could enhance the establishment of *Pseudomonas veronii* ptxD+ in a synthetic soil community for bioaugmentation. The engineered strain metabolized Phi, and its growth and toluene degradation were measured in different soil conditions. Phi boosted *P. veronii* ptxD+ abundance but only when phosphorus was limiting, with no added benefit when toluene was available. These findings show that while Phi can selectively support bioaugmentation, its effectiveness depends on the environmental context.

The study used a defined soil SynCom composed of 21 isolates to assess how Phi influences *Pseudomonas veronii* ptxD+ establishment with and without the added context of toluene degradation. Soil microcosms were prepared under different nutrient conditions, and Phi metabolism was tested by introducing the engineered strain. The authors relied heavily on 16S rRNA amplicon sequencing to track SynCom composition and shifts in community structure, providing a detailed view of microbial dynamics. However, most of these amplicon sequencing results were presented in supplementary figures, making it difficult to fully interpret their significance alongside the main results presented in the manuscript.

Overall, I found this study to be well designed and relatively novel. However, I think the authors can improve clarity and support the study's conclusions by moving some key supplementary figures into subpanels of the main figures; particularly those showing *P. veronii* ptxD+ relative abundance alongside SynCom members in the Grow and Stable conditions. Statistical

comparisons between +Phi and -Phi treatments at each time point should be applied to account for data spread and better quantify Phi's effect (see comments below). Finally, the Abstract should better emphasize one of the main results that Phi's effect is dependent on phosphorus limitation and does not enhance bioaugmentation when carbon is abundant, making its application highly context-dependent.

Other comments:

L94 the study's SynCom is reported to have "reproducible assembly and successional patterns". This should be cited or shown more rigorously in the Supplementary data. The level of experimental replication is not clear in Supplemental figures 2 & 3.

The opening paragraph of the Results section refers the reader to two supplemental figures. The authors might consider making these main figures given that the results were deemed important enough to lead the section off with. I think the results from this paragraph would be strengthened if the authors also measured the consumption rate or removal of total dissolved phosphorus between the wild type and *P. veronii* ptxD+ treatments along with the growth assay.

Supplemental Figures 9, 10 and 14 seem like they should be in the main text. I am curious why the relative abundance of the *P. veronii* ptxD+ strain was not shown at the species level in context with the rest of the SynCom for the Grow and Stable conditions. Given the use of a defined SynCom, this seems like key data/results to show in the respective time series measurements.

Figs 2F, 3H, 4C and 4D show a clear difference between +Phi and -Phi treatments when looking at trends shown by the averaged biological replicates. However, there is a notable spread of data that might obscure the results. I think this warrants some statistical analysis (e.g., Tukey test) to compare the means from -Phi and +Phi treatments within each time point to see if they are indeed significantly different from each other. It would probably be more thorough to do this for all the corresponding figures as well.

I am curious if the sequencing results showed amplification of non SynCom taxa that could have arisen from DNA left over in the sterile soil matrix? Are there any results that can or should be reported there?

Minor comments:

L74 and elsewhere refers to *Pseudomonas stutzeri* WM88. The authors may want to consider the change in of Genus to *Stutzerimonas* <https://doi.org/10.1016/j.syapm.2021.126289> <https://doi.org/10.3390/microorganisms10071363>

L451: is there a better way to reference the protocol than embedding the entire URL into the text?

Reviewer #3 (Comments for the Author):

Overall this paper was great to read and presented interesting results related to the enhanced degradation of toluene in the presence of Phi as a supplemental nutrient augmentation. The authors used a mix of experimental laboratory, metagenomic, and analytical chemistry techniques to quantify the degradation of toluene under several different conditions. The study used robust replication of microcosms (4 biological replicates) as well as positive and negative controls to isolate the specific variables of interest. This work does a wonderful job exploring how bioaugmentation remediation techniques can be instituted with Phi (phosphite) as a way to create a specialized niche for degrading organisms.

While the study is well structured and the analytical techniques are sound, I do have a few comments on the conclusions drawn from the study based on the study design. First, a major conclusion is that *P. veronii* ptxD+ responds more strongly to the presence of toluene and less strong to the Phi addition seems to be true, but there needs to be discussion on the preadaptation of the organism to toluene. This is described in the methods, but not in the paper and is important to understanding how *P. veronii* ptxD+ responded. Additionally, the "stable" microcosm were incubated for 7 weeks prior to inoculation with *P. veronii* ptxD+, however it was not discussed how the microcosms may have been anoxic/anaerobic upon inoculation. There is a clear difference in community composition data that is attributed to carbon depletion, but no discussion of the aerobic/anaerobic nature of the microcosms, which can also have a strong community selection effect. Additionally, the authors do discuss the role of remediation in a potentially carbon rich environment, which they do point out is different from the "stable" microcosms, but additional discussion of the oxygen dynamics and preadaptation is needed.

Line by Line comments are included below.

Specific comments:

Line 101: Degradation would be in soil not in soil extract. Recommend changing to "degradation in soil" or "toluene concentration in soil extracts".

Line 126 and Line 128: Why use 106-105 instead of something like "105 {plus minus} 102"? The figure is shown as an average with the individual measurements distributed, so it makes sense to have the text reflect this.

Line 127-128: Was the onset of "stable phase" determined by measuring cell density at a regular frequency until 106-105 CFU was measured? Is one measurement of 106-105 enough for the microcosm to be considered "stable"? Based on figure 2A, it seems like there was not continuous measurement of CFUs. While likely too late for this study, future work could use CFU

measurements to determine onset of stable or was there repeated measurements (i.e., 2-3 measurements of 106-105 CFU over one week to demonstrate stability).

Figure 2: I would put a D or D-F at the start of sentence "Absolutely *P. veronii* ptxD+..." to be consistent with the first half of the caption. Upon initial reading I thought the description applied to figure C. Keep (D), (E), (F) after specific test conditions for added clarity.

Figure 2: Why are there no data points between 5 days and 21 days for any measurements?

Line 148: Was the system carbon limited?

Line 164: What concentration solvent was used? Typically solvent addition to ecotoxicity studies should be < 0.1% solvent.

Figure 3: Same comment as Figure 2 about average and error bars, especially for the toluene concentrations.

Line 178: The claim that there was no change in cell counts over the 21-day period does not seem to match the data. The figure caption says the data is the same as 2B, which was described as "cell densities dropped steadily to 106-105 CFU/g soil (Figure 2B), which is consistent with previous observations." The cell density can't be decreasing in Figure 2 and constant in Figure 3.

Figure 4: Recommend changing X-axis for figures A and B to match C and D since there is only toluene data up to 70 hours. We have seen the 500 hour data in previous figures for CFUs.

Figure 5: At hour 0 in figure 5a and remove grey border box in figure 5B.

Line 231-232: Toluene consumption should not be written as a negative rate in this sentence. Rather, the consumption was "0.06 {plus minus} 0.03 h⁻¹" or you could say "Toluene concentration changed at a rate of -0.06 {plus minus} 0.03 h⁻¹ due to consumption."

Line 260-263: Where is the data for the Phi oxidation experiment? This seems to be missing from the SI, and if not available can references for other demonstrated cases of this be shared?

Line 378-380: It does not seem like the microcosms were aerated at all, which could contribute to the difference in community structure between the stable and grow conditions.

Line 404: It becomes clear in the methods that *P. veronii* ptxD+ was pre-adapted to toluene, which should be stated earlier in the paper. There needs to be additional discussion of the role of toluene pre-adaptation in toluene being a selective carbon substrate and supporting *P. veronii* ptxD+ responding strongly to the presence of toluene and less strong to the Phi addition. Upon revisiting figure 4, it seems very clear that the toluene presence/absence has a very strong effect on *P. veronii* ptxD+ establishment and clearance of toluene.

Other comments:

1. Please remove secondary x and y axes from plots unless absolutely needed. The excess tick marks clutter the figures.
2. Supplemental figure 4a: remove light grey box around legend
3. There should be at least one sentence that addresses the difference between metagenomics/a gene presence versus transcription and activity of a gene.

Reviewer #4 (Comments for the Author):

This manuscript presents an interesting contribution to the establishment of non-native organisms in native bacterial communities by taking a bioengineering approach along with the use of a synthetic soil community. One challenge to establishment into pre-existing communities can be nutrient limitations, such as phosphorous. The authors address this problem by engineering a toluene-degrading bacterial species to utilize phosphite as a source of phosphorous in order to overcome difficulties of establishment of non-native bacteria to into soil communities.

The authors demonstrate that toluene-degrading *Pseudomonas veronii* (PV) with the ptxD gene is capable of using phosphite as a phosphorus source and that it increased PV abundance in monoculture and in synthetic communities compared to WT. The authors also demonstrated that the presence of toluene does not affect abundance of syncom members and that the combination of phosphite and toluene does not alter PV ptxD+ abundance. While their hypothesis that phosphite would allow PV ptxD+ to form a niche in synthetic soil communities was not fully supported, their discovery that carbon (i.e. toluene) contributed to the niche formation is equally as important.

The studies presented were rigorously designed, interpretation was well supported, and the manuscript was clearly written. I commend the authors on their efforts in study design and analysis. Only a few minor suggestions are noted.

Minor comments:

1. Emphasis on utilizing a toluene-degrading organism was made, however, the effects of toluene and the importance its degradation were not made clear in the abstract/importance sections and was brief in the introduction. It may be useful to elaborate on the toxicity of this chemical to environmental systems.
2. Many figures are presented in hours but referred to in the text in days. It would be helpful to remain consistent and change the figure presentation to report days instead of hours.
3. Similarly, there are many references to changes in the community in "orders of magnitude", "fold change", "factor of 6", and "5x higher". Given the way the data is presented, it is sometimes difficult to see these changes on the graphs. Specifically Lines 143 - 150 referring to figure 2; lines 167-174 referring to figure 3.
4. Could the authors indicate how colony counting was performed (e.g. based on colony morphology or fluorescence). If colony counting was performed based on colony morphology, it would be more convincing to provide an image of the distinct colonies

for the 21-member synthetic community and PV.

5. Could the authors clarify how plasmid stability was assessed in PV for synthetic community studies? It is noted in the methodology that PV was kept under selective pressure until inoculated into the community. It would be nice to know that the plasmid is not lost during the length of the experiment without antibiotic pressure.

6. This work would benefit from inclusion of niche formation and succession invasion dynamics in the discussion.

Specific comments:

Lines 110-114 (results): Recommend moving the growth conditions and details in these lines to the methods section and briefly stating that the insertion was confirmed to be able to oxidize Phi.

Lines 121 - 123: Given the syncom was allowed to mature, it would be helpful to indicate that here by inclusion of "assembled" in the sentence "freshly diluted (assembled) SynCom or..."

Lines 138-139: It might be helpful to indicate what the authors are comparing PV ptxD+ growth to. Is it medium? Is it WT?

Lines 175 - 178: Similarly, these lines would benefit from clarifying what the community composition and diversity are being compared to.

Lines 188-190: Switching the reported percentage order of stable phase and grow phase to match the graph (grow, then stable) would assist the reader in following along with clarity.

Lines 194-198: It is not clear how carbon consumption is being measured and calculated.

Lines 215 - 217: It would be beneficial to have one ordination graph of the syncom showing +/- toluene, +/- PV, +/- Phi to visualize how each syncom clusters.

Line 263: It seems like there is a missing reference to a supplemental figure here.

Overall this paper was great to read and presented interesting results related to the enhanced degradation of toluene in the presence of Phi as a supplemental nutrient augmentation. The authors used a mix of experimental laboratory, metagenomic, and analytical chemistry techniques to quantify the degradation of toluene under several different conditions. The study used robust replication of microcosms (4 biological replicates) as well as positive and negative controls to isolate the specific variables of interest. This work does a wonderful job exploring how bioaugmentation remediation techniques can be instituted with Phi (phosphite) as a way to create a specialized niche for degrading organisms.

While the study is well structured and the analytical techniques are sound, I do have a few comments on the conclusions drawn from the study based on the study design. First, a major conclusion is that *P. veronii ptxD+* responds more strongly to the presence of toluene and less strong to the Phi addition seems to be true, but there needs to be discussion on the preadaptation of the organism to toluene. This is described in the methods, but not in the paper and is important to understanding how *P. veronii ptxD+* responded. Additionally, the “stable” microcosms were incubated for 7 weeks prior to inoculation with *P. veronii ptxD+*, however it was not discussed how the microcosms may have been anoxic/anaerobic upon inoculation. There is a clear difference in community composition data that is attributed to carbon depletion, but no discussion of the aerobic/anaerobic nature of the microcosms, which can also have a strong community selection effect. Additionally, the authors do discuss the role of remediation in a potentially carbon rich environment, which they do point out is different from the “stable” microcosms, but additional discussion of the oxygen dynamics and preadaptation is needed.

Specific comments:

Line 101: Degradation would be in soil not in soil extract. Recommend changing to “degradation in soil” or “toluene concentration in soil extracts”.

Line 126 and Line 128: Why use 10^6 - 10^5 instead of something like “ $10^5 \pm 10^2$ ”? The figure is shown as an average with the individual measurements distributed, so it makes sense to have the text reflect this.

Line 127-128: Was the onset of “stable phase” determined by measuring cell density at a regular frequency until 10^6 - 10^5 CFU was measured? Is one measurement of 10^6 - 10^5 enough for the microcosm to be considered “stable”? Based on figure 2A, it seems like there was not continuous measurement of CFUs. While likely too late for this study, future work could use CFU measurements to determine onset of stable or was there repeated measurements (i.e., 2-3 measurements of 10^6 - 10^5 CFU over one week to demonstrate stability).

Figure 2: I would put a *D* or *D-F* at the start of sentence “Absolutely *P. veronii ptxD+*...” to be consistent with the first half of the caption. Upon initial reading I thought the description applied to figure C. Keep (D), (E), (F) after specific test conditions for added clarity.

Figure 2: Why are there no data points between 5 days and 21 days for any measurements?

Line 148: Was the system carbon limited?

Line 164: What concentration solvent was used? Typically solvent addition to ecotoxicity studies should be < 0.1% solvent.

Figure 3: Same comment as Figure 2 about average and error bars, especially for the toluene concentrations.

Line 178: The claim that there was no change in cell counts over the 21-day period does not seem to match the data. The figure caption says the data is the same as 2B, which was described as “cell densities dropped steadily to 10^6 – 10^5 CFU/g soil (**Figure 2B**), which is consistent with previous observations.” The cell density can’t be decreasing in Figure 2 and constant in Figure 3.

Figure 4: Recommend changing X-axis for figures A and B to match C and D since there is only toluene data up to 70 hours. We have seen the 500 hour data in previous figures for CFUs.

Figure 5: At hour 0 in figure 5a and remove grey border box in figure 5B.

Line 231-232: Toluene consumption should not be written as a negative rate in this sentence. Rather, the consumption was “ $0.06 \pm 0.03 \text{ h}^{-1}$ ” or you could say “Toluene concentration changed at a rate of $-0.06 \pm 0.03 \text{ h}^{-1}$ due to consumption.”

Line 260-263: Where is the data for the Phi oxidation experiment? This seems to be missing from the SI, and if not available can references for other demonstrated cases of this be shared?

Line 378-380: It does not seem like the microcosms were aerated at all, which could contribute to the difference in community structure between the stable and grow conditions.

Line 404: It becomes clear in the methods that *P. veronii ptxD+* was pre-adapted to toluene, which should be stated earlier in the paper. There needs to be additional discussion of the role of toluene pre-adaptation in toluene being a selective carbon substrate and supporting *P. veronii ptxD+* responding strongly to the presence of toluene and less strong to the Phi addition. Upon revisiting figure 4, it seems very clear that the toluene presence/absence has a very strong effect on *P. veronii ptxD+* establishment and clearance of toluene.

Other comments:

1. Please remove secondary x and y axes from plots unless absolutely needed. The excess tick marks clutter the figures.
2. Supplemental figure 4a: remove light grey box around legend
3. There should be at least one sentence that addresses the difference between metagenomics/a gene presence versus transcription and activity of a gene.

Reviewer Response for “Phosphite as an Engineered Niche for *Pseudomonas veronii* in a Synthetic Soil Bacterial Community

Reviewer 1

We thank the reviewer for their positive feedback and suggested revisions. Several suggestions have been made by the reviewer (italics) to improve clarity, and are addressed below. Edits to the manuscript requested by Reviewer 1 are shown in red in the marked-up manuscript.

L109-110 "organism species" is awkward, I suggest revising to "taxonomic identity" here

The following edit has been made to the manuscript to improve readability:

Recombinant *P. veronii* colonies were verified with colony PCR to confirm the **taxonomic identity** and demonstrate the insertion of the *ptxD* gene (**Supplemental Figure 1A**).

*Suppl Fig 9 - the naming convention in the figure legend is inconsistent for *Pveronii* - the other organisms are all listed as genus names. Consider making it consistent or otherwise using correct notation (*P. veronii*).*

It would also be helpful to highlight the study organism in the legend perhaps in bold or with an "" to help readers quickly locate it.*

Check and do the same for other similar figures.

The label for *P. veronii ptxD*⁺ has been changed to “*P. veronii*” and put in bold text in the legends of Figure 3 and Supplemental Figures 9 and 14.

L149-150 - can you provide statistics to show if the 50 to 100-fold change is significant or not? It might not be. The stats provided on community shifts in the following sentences are helpful. Nonetheless, you should provide some stats here for the 50-100 percent statement, even if not significant.

In order to provide the suggested statistical analysis, the following additions were made to the manuscript:

Remarkably, addition of 12 µg Phi per g soil improves *P. veronii ptxD*⁺ competitiveness by a factor of 6 (**p = 0.035 by two-way ANOVA**), suggesting that Pi was one of the limiting factors for the competitive disadvantage of *P. veronii* (**Figure 2E**).

As expected, *P. veronii ptxD*⁺ was unable to invade a steady-state grown SynCom in the soil microcosms, since most available carbon has been depleted, but still addition of Phi enabled the inoculated strain to attain 50- to 100- fold higher abundances than in its absence (**p = 0.048, Figure 2F**).

L161 - modify to "...we were interested in whether..."

The following edit has been made to the manuscript:

Next, we were interested **in** whether the application of Phi could also improve the rates by which *P. veronii* would be able to degrade toluene in soil microcosms.

L164 - were any controls run to examine the effects of 2,2',4,4',6,8,8'-heptamethylnonane (HMN) alone?

There were no bottles containing only the carrier solvent in this experiment. HMN is considered generally microbially inert due to its saturated alkane structure and its use is well-precedented in the literature. However, it is possible that the presence of HMN facilitated the extraction of certain nutrients or inhibitors from the soil matrix that would be otherwise unavailable to the SynCom and/or inoculant. We thank the reviewer for this point and will incorporate these controls in future experiments.

L166-168 - this sentence is confusing, especially what the "its" means in "in its absence". Clarify if you're referring to the absence of toluene, P veronii, the SynCom, etc.

The following revision has been made to the manuscript to improve clarity:

In the absence of the SynCom, but in presence of toluene, *P. veronii ptxD*⁺ growth was 5 times higher (**Figure 3A**) **than in toluene-free microcosms (Figure 2D)**, indicating utilization of toluene (**Figure 3B**).

L167-174 - In this passage, there are many statements here about differences in growth, toluene clearance rate, etc. but there are no stats presented. The figures (Fig. 3) are compelling, but statistical analyses are also needed in order to claim that these trends are truly different. Also, be cautious about the term "rate" since that's a specific measurement that is expressed in units of toluene over time, vs. just a trend of levels over time. You may want to just discuss toluene loss (rather than rate) if you did not calculate rates here.

We thank the reviewer for this suggestion of how to strengthen our claims. Statistical analysis using two-way ANOVA was used to evaluate the statements in the following passage:

Toluene loss (Figure 3B) was significantly greater than the evaporative loss in sterile controls (**p = 0.020, by two-way ANOVA, Figure 3C**), which retain over 40% of the initial toluene present after 8 days. The addition of Phi did not affect growth (**p = 0.40**) or toluene (**p = 0.63**) clearance in *P. veronii ptxD*⁺ mono-colonization of the soil microcosms (**Figure 3A, B**). The total SynCom growth without inoculated *P. veronii* after mixing with fresh soil matrix (grow phase) containing

130 µg toluene per g soil was indistinguishable from growth in soil without toluene or HMN ($p = 0.34$, **Figure 3D**), and toluene loads as high 865 µg/g soil could be tolerated as long as HMN was used as the delivery mechanism (**Supplemental Figure 5**)...

Toluene exposure during the stable phase also showed no change in total cell counts over the course of the 21-day observation period ($p = 0.57$, by two-way ANOVA **Figure 3G**) and a decrease in richness ($p = 4.83 \times 10^{-8}$, **Supplemental Figure 7B**).

These values have also been added to the figure captions.

L278 - "contamination sites" should be "contaminated sites"

The following edit has been made to the manuscript:

The larger selective effect of toluene in comparison to Phi suggests that the soil microcosm model used in this study is C-limited, rather than P-limited, which may not be representative of potential **contaminated** sites (36, 37).

Reviewer 2

We appreciate the reviewer's encouraging comments and their suggested revisions to improve clarity. We have addressed their questions and suggestions (shown in italics) below, with changes to the manuscript in response to Reviewer 2 are shown in **blue**.

Finally, the Abstract should better emphasize one of the main results that Phi's effect is dependent on phosphorus limitation and does not enhance bioaugmentation when carbon is abundant, making its application highly context-dependent.

The following edit has been made to the manuscript abstract to address this important conclusion from our study:

When inoculated in either soil matrix or liquid soil extract, *P. veronii* proliferates in a phosphite- and toluene-dependent manner in both growing and stable synthetic soil microbial communities, **although the selective effects of phosphite and toluene were not additive in a carbon-limited context.**

L94 the study's SynCom is reported to have "reproducible assembly and successional patterns". This should be cited or shown more rigorously in the Supplementary data. The level of experimental replication is not clear in Supplemental figures 2 & 3.

We appreciate the reviewer bringing to our attention to the apparent lack of evidence for this claim. This statement was supported through the citation of the previous sentence, which references *Reproducible Propagation of Species-Rich Soil Bacterial Communities Suggests Robust Underlying Deterministic Principles of Community Formation* by Čaušević *et al.*, where

the assembly and succession patterns of the SynCom were studied extensively. To make this clearer, we have moved the citation as shown below:

To assess the effect of Phi on the establishment of the *P. veronii ptxD*⁺ inoculant and toluene degradation rates, we use a model synthetic soil community (SynCom) in soil microcosms as our background. This community consists of 21 soil isolates (**Supplemental Table 1**) covering four major bacterial phyla that have reproducible assembly and successional patterns (23).

*The opening paragraph of the Results section refers the reader to two supplemental figures. The authors might consider making these main figures given that the results were deemed important enough to lead the section off with. I think the results from this paragraph would be strengthened if the authors also measured the consumption rate or removal of total dissolved phosphorus between the wild type and *P. veronii ptxD*⁺ treatments along with the growth assay.*

We thank the reviewer for this suggestion. While the successful insertion of the *ptxD* into the genome of *P. veronii* is essential to establish before further discussion, we do not consider this to be a primary result of our study. Since we consider this data more so as a validation of the biological material used in the experiment, we placed it in the supplemental figure section.

We appreciate the reviewer's suggestion to incorporate P assimilation rates in our comparison of the *ptxD*⁺ mutant and the wild type. While we will investigate this and incorporate it into further studies, we do not believe that these consumption rates would necessarily be relevant to how the inoculant assimilates Phi in the soil microcosm, where [Phi] ~ [Pi] and Pi is bound to solid soil matrix.

Supplemental Figures 9, 10 and 14 seem like they should be in the main text.

While we agree that the stacked bar plots present a helpful overview of the SynCom or SynCom + inoculant, we opted to extract the most relevant information (e.g. the abundance of *P. veronii* compared between two conditions) in order to help communicate the main results to the reader in the clearest manner for the general readership of mSystems. However, we agree that for certain key results this is important to include in the main text (see below).

*I am curious why the relative abundance of the *P. veronii ptxD*⁺ strain was not shown at the species level in context with the rest of the SynCom for the Grow and Stable conditions. Given the use of a defined SynCom, this seems like key data/results to show in the respective time series measurements.*

We apologize if this was confusing. The relative abundances of *P. veronii* in toluene-contaminated conditions, with Phi or not, were shown in the Supplemental Figure S10. Upon suggestion of the reviewer, we have merged Supplemental Figure S10 and the stacked relative abundances in the absence of toluene into a new Fig. 3.

Figs 2F, 3H, 4C and 4D show a clear difference between +Phi and -Phi treatments when looking

at trends shown by the averaged biological replicates. However, there is a notable spread of data that might obscure the results. I think this warrants some statistical analysis (e.g., Tukey test) to compare the means from -Phi and +Phi treatments within each time point to see if they are indeed significantly different from each other. It would probably be more thorough to do this for all the corresponding figures as well.

We thank the reviewer for this suggestion, and have received a similar piece of feedback from Reviewer 1. We have done statistical analysis by two-way ANOVA and added corresponding p-values to our claims in the manuscript and is detailed above under “Reviewer 1”. In order to add statistical backing to our analysis of Figure 4C and 4D as mentioned by the reviewer, the following edit has been made to the manuscript:

No Phi effect was observed for toluene clearance, which was complete within the first 48 h in both grow and stable phases with or without Phi (**Figure 4C and D**, $p = 0.42$ and 0.16 for grow and stable phase respectively, of timepoints 0 h to 48 h, by two-way ANOVA).

These values have also been added to the figure caption.

I am curious if the sequencing results showed amplification of non SynCom taxa that could have arisen from DNA left over in the sterile soil matrix? Are there any results that can or should be reported there?

In our procedure the isolation of relic DNA from soil is minimized, because we first wash cells from the soil matrix and only then purify DNA from the collected cells (lines 408-419 in the Materials and Methods section). Of all of the soil microcosm samples that were sequenced, $70 \pm 4\%$ of the total read counts were attributed to a member of SynCom. However, this does not necessarily indicate that the remaining read counts were from “foreign” DNA, but potentially from sequences that were trimmed too short to match the search sequence or have small misreads in sequences that were not mapped to any SynCom member with the “grep” function.

*L74 and elsewhere refers to *Pseudomonas stutzeri* WM88. The authors may want to consider the change in of Genus to *Stutzerimonas* <https://doi.org/10.1016/j.syapm.2021.126289> <https://doi.org/10.3390/microorganisms10071363>*

We thank the reviewer for bringing this to our attention. The following edit has been made to the manuscript to reflect the new classification:

We have previously shown that, when applied to soil, the phosphite oxidizing bacterium *Pseudomonas stutzeri* WM88 (now reclassified as *Stutzerimonas* (13,14)) is capable of Phi-dependent growth and can oxidize sufficient Phi to support plant growth...

L451: is there a better way to reference the protocol than embedding the entire URL into the text?

We will inquire with an editor at *mSystems* if it is possible to embed a hyperlink in order to improve readability.

Reviewer 3

We thank the reviewer for their enthusiasm and positive feedback, as well as their insightful comments and suggestions. Their suggestions, edits, and questions (in italics) are addressed below. Revisions to the manuscript based on Reviewer 3's comments are shown in orange.

*First, a major conclusion is that *P. veronii* ptxD+ responds more strongly to the presence of toluene and less strong to the Phi addition seems to be true, but there needs to be discussion on the preadaptation of the organism to toluene. This is described in the methods, but not in the paper and is important to understanding how *P. veronii* ptxD+ responded.*

We thank the reviewer for this suggestion. We did not pre-adapt *P. veronii* to growth on toluene. As mentioned in lines 383-385, the preculture of *P. veronii* before inoculation was raised with succinate. Toluene selection was only applied directly after plating –80 stocks to ensure that the 'chromosome 2'-DNA with the toluene degradation genes (*ibp*, *dmp*, and *ttg*) remained present in the strain. We added this to the discussion in lines 304-306 (with the relevant reference), and to the materials section in line 297-299.

Toluene degradation in *P. veronii* 1YdBTEX2 is mediated by the products of the *ibp*, *dmp* and *ttg* genes, which are all located on the chromosome-2-replicon and strongly induced in presence of BTEX (39).

Inoculants were thus not pre-adapted to toluene before their addition to the soil microcosms.

*Additionally, the "stable" microcosm were incubated for 7 weeks prior to inoculation with *P. veronii* ptxD+, however it was not discussed how the microcosms may have been anoxic/anaerobic upon inoculation. There is a clear difference in community composition data that is attributed to carbon depletion, but no discussion of the aerobic/anaerobic nature of the microcosms, which can also have a strong community selection effect. Additionally, the authors do discuss the role of remediation in a potentially carbon rich environment, which they do point out is different from the "stable" microcosms, but additional discussion of the oxygen dynamics and preadaptation is needed.*

We thank the reviewer for bringing up this point. Indeed, we did not specifically use soil microcosms with aerobic/anaerobic conditions because that is much more difficult to control in batch flasks since we would need a constant water table, and can only be reasonably carried out in lysimeter or column studies. However, we did previously show that *P. veronii* is capable of growth under anoxic conditions – although this is in itself incompatible with BTEX degradation, since this is carried out by dioxygenase enzymes (IbpAaAbAcAd).

We inserted a small text paragraph at line 302 and referred to this work: 10.1371/journal.pone.0165850.g003.

Although for simplicity of the study we focused here entirely on simulating the effects of aerobic soil conditions, real contaminated environments pose the additional complexity of alternating oxic/anoxic conditions and/or water-saturated zones. It is interesting that *P. veronii* 1YdBTEX2 is capable of growth under denitrifying conditions and could thus potentially survive or proliferate during periods of anoxic conditions provided that nitrite or nitrate are available (40). Anoxic conditions themselves, though, are incompatible with BTEX degradation by *P. veronii* 1YdBTEX2, since the degradation pathway is mediated by dioxygenase enzymes (IbpAaAbAcAd).

Line 378-380: It does not seem like the microcosms were aerated at all, which could contribute to the difference in community structure between the stable and grow conditions.

We thank the reviewer for bringing this to our attention. The Schott glass bottles we used in the experiment are filled with 100 g soil, maintained at 10% gravimetric water content and with an air volume of 400 ml, which is sufficient to keep aerobic conditions (micro-anoxic pockets may be formed, though). In addition, we place the flasks on a horizontal roller before each sampling to ensure some mixing. Therefore, and as we have shown previously, we don't believe that differences in aerobicity or oxygen provision have caused differences in community structure between stable and grow condition. Rather, substrate availability was the cause, as can also be seen from the general increase in community size (Fig. 2A). No further changes are needed.

Line 101: Degradation would be in soil not in soil extract. Recommend changing to "degradation in soil" or "toluene concentration in soil extracts".

We apologize for this confusion here. "Degradation in soil extracts" referred to the liquid culture experiments. We changed line 108/109 as follows:

..contaminated soil ~~and soil extract~~, and enhances toluene degradation **in liquid soil extract cultures.**

Line 126 and Line 128: Why use 106-105 instead of something like "105 {plus minus} 102"? The figure is shown as an average with the individual measurements distributed, so it makes sense to have the text reflect this.

We thank the reviewer for this remark. The issue is that log-scale variations are always difficult to convey, which is why we believe that it is easier to comprehend a range of log-values (10e5-10e6) instead of the mean log-value and its standard error (e.g., 10e8.3 ± 10e0.13). We prefer to keep this as is.

Line 127-128: Was the onset of "stable phase" determined by measuring cell density at a regular frequency until 10⁶-10⁵ CFU was measured? Is one measurement of 10⁶-10⁵ enough for the microcosm to be considered "stable"? Based on figure 2A, it seems like there was not continuous measurement of CFUs. While likely too late for this study, future work could use CFU measurements to determine onset of stable or was there repeated measurements (i.e., 2-3 measurements of 10⁶-10⁵ CFU over one week to demonstrate stability.

We thank the reviewer for this suggestion to improve our establishment of a “stable phase”. Indeed, the cell densities were not measured continuously throughout the maturation of SynCom in soil microcosm. The length of incubation for “stable phase” was determined based on previous results (see Čaušević *et al.*, 2022, ref. 23) which showed that a stable community composition with restored richness was achieved after prolonged incubation of 2-6 months. However, stable is only used in a relative and operational sense, because slower cell death and turnover by predation (e.g., by the opportunistic predator *Lysobacter*) will continue to act on the SynCom community. For the point we are trying to make here, that the major fraction of available carbon is consumed within the first two weeks, and therefore, there is a difference between a ‘stable’ and a ‘grow’ community, our measurements suffice. No further changes needed.

Figure 2: I would put a D or D-F at the start of sentence "Absolutely P. veronii ptxD+..." to be consistent with the first half of the caption. Upon initial reading I thought the description applied to figure C. Keep (D), (E), (F) after specific test conditions for added clarity.

We thank the reviewer for bringing this to our attention. To improve clarity of the figure caption, the following edit has been made:

D–E Absolute *P. veronii* ptxD⁺ abundance over time in soil microcosm in the presence (dark aqua diamonds), and absence (dark aqua circles) of 12 µg Phi per g soil when inoculated in soil microcosm alone (**D**), in the presence of growing SynCom (**E**), and in stable, grown SynCom (**F**).

Figure 2: Why are there no data points between 5 days and 21 days for any measurements?

Figure 3: Same comment as Figure 2 about average and error bars, especially for the toluene concentrations.

We aimed to use our experimental resources as efficiently as possible in an experiment at this scale, while still capturing the most dynamic moments of SynCom and inoculant growth and development. Accordingly, we chose to sample at a higher resolution in the first 48 h, capturing the exponential phase of growth, the peak cell densities, and the period of toluene clearance (in relevant experiments). The remaining data points at 7/8 days capture “early stationary” phase and the data points at 21 days capture “late stationary” phase.

Line 148: Was the system carbon limited?

Due to the dramatic effect that the addition of a selective carbon source (toluene) had on community composition (Supplemental Figure 9), we infer that the system is carbon limited.

Line 164: What concentration solvent was used? Typically solvent addition to ecotoxicity studies should be < 0.1% solvent.

The carrier solvent was added at a concentration of 2% v/v to the soil extract (L386) before addition to the soil matrix, with a final gravimetric concentration of 0.2%. While this is slightly higher than the reviewer's recommended solvent concentration, the carrier solvent was chosen due to its inertness to microbial activity. The concentrations of toluene, however, are much lower in the soil microcosm, with a final gravimetric concentration of 0.015%.

Line 178: The claim that there was no change in cell counts over the 21-day period does not seem to match the data. The figure caption says the data is the same as 2B, which was described as "cell densities dropped steadily to 10⁶-10⁵ CFU/g soil (Figure 2B), which is consistent with previous observations." The cell density can't be decreasing in Figure 2 and constant in Figure 3.

In the following sentence in the manuscript:

Toluene exposure during the stable phase also showed no change in total cell counts over the course of the 21-day observation period (**Figure 3G**) and a decrease in richness ($p = 4.83 \times 10^{-8}$, **Supplemental Figure 7B**).

The “no change” is referring to the comparison of cell counts between toluene-contaminated and toluene-free soil microcosm shown in Figure 3G. It is true in both figures that the cell counts are decreasing over time. To avoid confusion for the reader, we have made the following edit to the manuscript for clarity:

Toluene exposure during the stable phase **had no effect on** total cell counts over the course of the 21-day observation period ($p = 0.57$, by two-way ANOVA **Figure 3G**) and a decrease in richness ($p = 4.83 \times 10^{-8}$, **Supplemental Figure 7B**).

Figure 4: Recommend changing X-axis for figures A and B to match C and D since there is only toluene data up to 70 hours. We have seen the 500 hour data in previous figures for CFUs.

We thank the reviewer for this suggestion. Although we agree it is not optimal to have different axes for A/B and C/D, we have opted to use a shorter time window for the toluene data. Since toluene is completely consumed within 48 h, a x-axis scale that extends to 500 h would be mostly blank space—this truncated window allows us to see the “interesting” part of the data in higher detail, which includes the period of toluene consumption in the 48 h and the demonstration of consistently ~ 0 $\mu\text{g/g}$ soil measurement in the following two timepoints.

Figure 5: At hour 0 in figure 5a and remove grey border box in figure 5B.

We thank the reviewer for bringing this to our attention, and Figure 5 has been fixed accordingly.

Line 231-232: Toluene consumption should not be written as a negative rate in this sentence. Rather, the consumption was "0.06 {plus minus} 0.03 h⁻¹" or you could say "Toluene concentration changed at a rate of -0.06 {plus minus} 0.03 h⁻¹ due to consumption."

The following modification has been made to the manuscript for accuracy:

Using an exponential fit of the data, toluene consumption was $0.06 \pm 0.03 \text{ h}^{-1}$ in the presence of Phi and $0.04 \pm 0.01 \text{ h}^{-1}$ in the absence of Phi.

Line 260-263: Where is the data for the Phi oxidation experiment? This seems to be missing from the SI, and if not available can references for other demonstrated cases of this be shared?

We thank the reviewer for bringing this to our attention. Since we do not have the data to support this statement readily available, we have removed the statements referencing the Phi oxidation experiment.

~~To determine if these strains were capable of Phi oxidation, individual members were inoculated in 5 mL 21C liquid media prepared with Phi replacing Pi as the phosphorus source (23 mM Phi) and buffer system at pH 6.8 and 10 mM succinate as the carbon (C) source. Of these three enriched strains, only Chitinophaga and Luteibacter were capable of growth on Phi.~~

Line 404: It becomes clear in the methods that *P. veronii* ptxD⁺ was pre-adapted to toluene, which should be stated earlier in the paper. There needs to be additional discussion of the role of toluene pre-adaptation in toluene being a selective carbon substrate and supporting *P. veronii* ptxD⁺ responding strongly to the presence of toluene and less strong to the Phi addition. Upon revisiting figure 4, it seems very clear that the toluene presence/absence has a very strong effect on *P. veronii* ptxD⁺ establishment and clearance of toluene.

We apologize for this confusion, but as we mention above in reply to the more general question, *P. veronii* was NOT pre-adapted to toluene before inoculation. The preculture was raised in 15 mM succinate (see lines 383-385). But indeed, as we also mention in our reply above and the corresponding text revisions, toluene degradation is inducible in *P. veronii* 1YdBTEX2.

Please remove secondary x and y axes from plots unless absolutely needed. The excess tick marks clutter the figures.

The secondary axes are provided for precision of evaluation of the data in printed form. Given the value added with the secondary axes and the subjective nature of the critique, we respectfully prefer to maintain the figures in their current form.

2. Supplemental figure 4a: remove light grey box around legend

Supplemental Figure 4 has been updated with this box removed.

3. There should be at least one sentence that addresses the difference between metagenomics/a gene presence versus transcription and activity of a gene.

We thank the reviewer for this suggestion, and have added the following statement to the manuscript at Line 264:

Whether *phoV* is not only present in the genome of these organisms but also expressed and active as PhoV is an area of future study.

Reviewer 4

We thank the reviewer for their response and enthusiasm for this work. The reviewer has presented some aspects of the study design and analysis which require further explanation, and provided some suggestions to improve the clarity of the manuscript. These comments are presented in italics and the author response is shown below each comment. Revisions made to the manuscript based on Reviewer 4's responses are shown in green.

Minor comments:

1. Emphasis on utilizing a toluene-degrading organism was made, however, the effects of toluene and the importance its degradation were not made clear in the abstract/importance sections and was brief in the introduction. It may be useful to elaborate on the toxicity of this chemical to environmental systems.

We did not greatly elaborate on the effects of toluene since we wanted to emphasize the broader applicability of this study to inoculant establishment and microbiome engineering in general. However, we agree that this is a missing component of the abstract and have made the following addition to the manuscript:

Bioaugmentation, the process of soil restoration by introducing microorganisms capable of degrading pollutants, is a promising and cost-effective strategy for environmental remediation. Aromatic hydrocarbons, such as benzene, toluene, ethylbenzene and *p*-xylene (BTEX), are highly toxic environmental contaminants that could be transformed to less harmful products through the inoculation of certain organisms capable of BTEX degradation. However, a barrier to successful bioaugmentation...

2. Many figures are presented in hours but referred to in the text in days. It would be helpful to remain consistent and change the figure presentation to report days instead of hours.

The following adjustments have been made to the manuscript to create agreement between the hour scale used in figures and the discussion in the text:

Toluene loss (**Figure 3B**) was significantly greater than the evaporative loss in sterile controls ($p = 0.020$, by two-way ANOVA, **Figure 3C**), which retain over 40% of the initial toluene present after 192 h (8 d).

Toluene measurements also indicated a slightly faster removal when *P. veronii* was inoculated, allowing the toluene to reach < 0.6% of the initial concentration after 48 h (**Figure 3E, Figure 3I**).

Timepoints for the growing phase were taken 24 h, 48 h, 96 h (4 d), 168 h (7 d), and 504 h (3 wk) after inoculation. Timepoints for the stable phase were taken 24 h, 48 h, 120 h (5 d), 192 h (8 d, except for the SynCom-only control), and 504 h (3 wk).

3. Similarly, there are many references to changes in the community in "orders of magnitude", "fold change", "factor of 6", and "5x higher". Given the way the data is presented, it is sometimes difficult to see these changes on the graphs. Specifically Lines 143 - 150 referring to figure 2; lines 167-174 referring to figure 3.

We agree with the reviewer that visualizing the fold-changes in abundance that are reported in the text on the log-scale plots is not obvious. Unfortunately we find this to be preferable to reporting CFU values on a non-log scale, or reporting increases in *P. veronii* abundance in fractions of log values.

4. Could the authors indicate how colony counting was performed (e.g. based on colony morphology or fluorescence). If colony counting was performed based on colony morphology, it would be more convincing to provide an image of the distinct colonies for the 21-member synthetic community and PV.

Colony counting was performed on R2A based on colony morphology, but individual SynCom members were not distinguished and only the total CFU/g soil was recorded. These values were adjusted for relative abundance of each member using the 16S rRNA gene amplicon sequencing. Importantly, both counting and DNA isolation + amplicon sequencing were done on the same washed cell suspension from the soil samples. Therefore, the quantification of individual SynCom population sizes by multiplying the total community size (CFU counts) with the relative abundances from sequence reads (corrected for the genome copies) is justified.

To clarify the method of colony counting, images of the colonies growing on R2A agar plates are shown in a new Supplemental Figure 14, which is referenced under "Materials and Methods".

The SynCom was assembled from plates (**Supplemental Figure 14**) of the 21 isolates individually as described previously (23).

5. *Could the authors clarify how plasmid stability was assessed in PV for synthetic community studies? It is noted in the methodology that PV was kept under selective pressure until inoculated into the community. It would be nice to know that the plasmid is not lost during the length of the experiment without antibiotic pressure.*

The mini-Tn7 system does not involve plasmid maintenance in the cell, but instead chromosomal insertion downstream of the highly conserved *glmS* genes found in many bacteria. Since it is delivered in a suicide plasmid which cannot sustain replication in the cell, we can assume that the originally propagated *ptxD*⁺ mutant maintains its Phi-oxidizing functionality in the experiment.

6. *This work would benefit from inclusion of niche formation and succession invasion dynamics in the discussion.*

We thank the reviewer for the remark. This topic was discussed in lines 306-315 in the context of cross-feeding.

Specific comments:

Lines 110-114 (results): Recommend moving the growth conditions and details in these lines to the methods section and briefly stating that the insertion was confirmed to able to oxidize Phi.

The following revisions have been made to the manuscript:

[In “Results”]

Recombinant *P. veronii* colonies were verified with colony PCR to confirm the **taxonomic identity** and demonstrate the insertion of the *ptxD* gene (**Supplemental Figure 1A**). To verify that the *ptxD* insertion conveyed the functional ability to oxidize Phi, *P. veronii* WT and *P. veronii ptxD*⁺ were inoculated Phi- or Pi-containing 21C media. While both the WT and *ptxD*⁺ strains were capable of growth in Pi medium, only *P. veronii ptxD*⁺ displayed growth in the Phi medium (**Supplemental Figure 1B**).

[In “Methods”]

To verify Phi oxidizing activity, 21C liquid media (3) was prepared with Phi replacing Pi as the phosphorus source (23 mM Phi) and buffer system at pH 6.8 and 10 mM succinate as the carbon source. *P. veronii* 1YdBTEX2 WT or *ptxD*⁺ colonies were inoculated into 5 mL of Phi- or Pi-containing 21C media. After incubating at 30°C for 2 days, cultures were checked for growth by their culture turbidity.

Lines 121 - 123: Given the syncom was allow to mature, it would be helpful to indicate that here by inclusion of "assembled" in the sentence "freshly diluted (assembled) SynCom or..."

We thank the reviewer for this suggestion to improve clarity. The following edit has been made to the manuscript:

To evaluate the efficacy of the expressed recombinant *ptxD* gene in *P. veronii* to aid in proliferation in a Phi-amended soil within a microbial consortium background, we inoculated the strain at 10^5 cells per gram either concomitantly with freshly diluted, **assembled** SynCom or after full growth of the SynCom in soil microcosms.

Lines 138-139: It might be helpful to indicate what the authors are comparing PV ptxD+ growth to. Is it medium? Is it WT?

We apologize if this caused confusion. We introduce the topic here in lines 123-125: the extent of differences in growth within the SynCom under GROW conditions (i.e., available nutrient niches for all strains simultaneously, hence competition) and STABLE (expected available nutrient niches to be depleted by the SynCom members). We think the confusion stems from not explicitly mentioning that we are looking here first at the SynCom development without *P. veronii*. This was now added to lines 122-124 and the caption for Figure 2.

..or after full growth of the SynCom in soil microcosms. **Assembled SynComs in absence of *P. veronii***, in the "grow" stage with ~~the transfer of assembled SynCom to~~ fresh soil extract (SE) and soil matrix (SM) **have sufficient available carbon to** allowed the community to **increase in size and** reach maximum cell..

[Figure 2 caption]

A Total colony forming units (CFU) per g soil over time of the SynCom **without *P. veronii* ptxD⁺** in the grow phase. **B** Total colony forming units (CFU) per g soil over time of the SynCom **without *P. veronii* ptxD⁺** in the stable phase.

Lines 175 - 178: Similarly, these lines would benefit from clarifying what the community composition and diversity are being compared to.

Here the SynCom composition in a toluene-contaminated microcosm is being compared to SynCom composition in a toluene-free microcosm. To make this more clear, the following edit has been made to the manuscript:

The community composition (**Supplemental Figure 6**) and alpha diversity analysis showed a significant change in richness ($p = 0.026$, Welch's t-test), but no change in the Shannon or Simpson diversity indices **in comparison to the toluene-free control (Supplemental Figure 7A)**.

Lines 188-190: Switching the reported percentage order of stable phase and grow phrase to match the graph (grow, then stable) would assist the reader in following along with clarity.

The following edit has been made to the manuscript to maintain consistency:

The inoculant contributed up to 46% of the total community counts in the grow phase and up to 81% in the stable phase of SynCom (Supplemental Figure 9).

Lines 194-198: It is not clear how carbon consumption is being measured and calculated.

Carbon consumption was calculated from net toluene loss by firstly subtracting the toluene remaining in the + *P. veronii* soil microcosm from the toluene contained in the – *P. veronii* soil microcosm at the 48 h timepoint.

e.g. 64.6 µg tol/g soil – 0.24 µg tol/g soil = 64.4 µg tol/ g soil

This was then converted to g C/g soil using the molecular weight of toluene (92 g/mol), and the weight of carbon in the molecule (84 g/mol).

e.g. $\frac{84 \frac{\text{g C}}{\text{mol}}}{92 \frac{\text{g tol}}{\text{mol}}} \times (6.4 \times 10^{-5} \text{ g tol/g soil}) = 5.9 \times 10^{-5} \text{ g C/g soil}$

Lines 215 - 217: It would be beneficial to have one ordination graph of the syncom showing +/- toluene, +/- PV, +/- Phi to visualize how each syncom clusters.

We thank the reviewer for this suggestion, and have shown the NMDS ordination for all conditions contained in one plot below. However, we find that the plot is too busy for meaningful interpretation by the reader, and for this reason have favored smaller subsets of conditions for the NMDS ordinations in this manuscript.

Line 263: It seems like there is a missing reference to a supplemental figure here.

We thank the reviewer for this remark. This was also brought to our attention by Reviewer 3 and our response and manuscript edits can be seen under “Reviewer 3”.

Other revisions

The sections “Construction of *P. veronii ptxD*⁺”, “Community 16S rRNA Gene Amplicon Sequencing”, and “Toluene Quantification by GC-MS” from Materials and Methods have been moved to the Supplemental Information document to adhere to the mSystems word limit.

The following revisions were made to the manuscript to correct an error in statistical analysis and are shown in purple.

In presence of toluene, the competitive advantage of *P. veronii ptxD*⁺ within the soil SynCom was again increased, as expected from toluene being a selective carbon substrate, and population densities were enriched up to 10-fold (in growing SynCom background, Day 4) and 1000-fold (in stable SynCom, Day 2; **Figure 3E** and **3H**, $p = 4.1 \times 10^{-4}$ and 2.6×10^{-10} by two-way ANOVA for grow and stable phase, respectively).

Indeed, *P. veronii ptxD*⁺ exhibited higher absolute abundances in toluene-contaminated SE medium in the presence of Phi for the first 48 h after inoculation than in its absence (**Figure 5A**, $p = 9.1 \times 10^{-5}$, two-way ANOVA).

The composition of the SynCom in the presence of Phi was also significantly changed by the presence of *P. veronii ptxD*⁺ according to the NMDS ordination ($p = 0.0040$ and 0.0020 for grow and stable phase, respectively, by two-way ANOVA) in comparison to the absence of Phi **in grow phase** ($p = 0.185$), **but similarly to the absence of Phi in stable phase** ($p = 0.003$), indicating community response to the competitive advantage imparted to the inoculant by Phi **at the growth levels attained from fresh substrate** (**Supplemental Figure 4**).

Re: mSystems00061-25R1 (Phosphite as an Engineered Niche for *Pseudomonas veronii* in a Synthetic Soil Bacterial Community)

Dear Dr. Clara Bailey:

The expert reviewers and I are in agreement that the revisions made have mostly addressed the original concerns. However, there were a few cases where more changes and clarifications were warranted. Specifically, reviewer 2 brought up concerns about relic DNA, and the authors implied that 30% of the reads were not aligned to the SynCom members. This should be addressed in both the Methods and Results, and some clarity about the composition of these remaining 30% of reads should certainly be presented in the data set. There was also a comment about referencing methods/protocols by embedding a URL. Please be more robust in your citation here by either providing a scientific reference for the material and/or detailing methodological steps performed. The URL is not sufficient.

Reviewer 3 had concerns about the author's rationale for absolute total abundance. I also encourage the authors to make specific revisions to the paper to better satisfy these concerns.

At this stage, I think it is most appropriate for the authors to address all of the remaining concerns by making changes rather than rebuttals, unless there is a strong argument otherwise. This should include justification of data needed to interpret main results being located in the supplementary files. In general, if the data is needed to support the results, then it should be included in one of the main figures.

Revision Guidelines

Sincerely,
Hans Bernstein
Editor
mSystems

Reviewer #2 (Comments for the Author):

Round 2 Review for "Phosphite as an Engineered Niche for *Pseudomonas veronii* in a Synthetic Soil Bacterial Community"

The authors addressed a portion of my previous comments and in most cases made changes to the manuscript. In this round of review, I am listing the original comments that were not addressed and, in some cases, giving additional feedback on the author's responses.

As stated in my first review, the authors relied heavily on 16S rRNA amplicon sequencing to track SynCom composition and shifts in community structure, providing a detailed view of microbial dynamics. However, most of these amplicon sequencing results were presented in supplementary figures, making it difficult to fully interpret their significance alongside the main results presented in the manuscript. I suggested that authors should improve clarity and support the study's conclusions by moving some key supplementary figures into subpanels of the main figures; particularly those showing *P. veronii* ptxD+ relative abundance alongside SynCom members in the Grow and Stable conditions.

This was not really addressed. I see that there is an updated Figure 3, but the main text frequently requires the reader to go into the supplementary information to validate main results of the paper. I highly suggest that the authors do a more careful screening on what data is needed to support the results and show this as a subpanel or full figure.

Other comments:

L94 the study's SynCom is reported to have "reproducible assembly and successional patterns". This should be cited or shown more rigorously in the Supplementary data. The level of experimental replication is not clear in Supplemental figures 2 & 3.

Author's response: "We appreciate the reviewer bringing to our attention to the apparent lack of evidence for this claim. This statement was supported through the citation of the previous sentence, which references Reproducible Propagation of Species-Rich Soil Bacterial Communities Suggests Robust Underlying Deterministic Principles of Community Formation by Čaušević et al., where the assembly and succession patterns of the SynCom were studied extensively. To make this clearer, we have moved the citation as shown below:

To assess the effect of Phi on the establishment of the *P. veronii* ptxD+ inoculant and toluene degradation rates, we use a model synthetic soil community (SynCom) in soil microcosms as our background. This community consists of 21 soil isolates (Supplemental Table 1) covering four major bacterial phyla that have reproducible assembly and successional patterns (23)."

I do not see where this citation specifically applies to the system studied here. I don't think its appropriate to simply generalize that soil community succession and assembly is deterministic and not context dependent. I suggest to either eliminate this claim or do analyses that would be included in the main text that support this as a result .

I am curious if the sequencing results showed amplification of non SynCom taxa that could have arised from DNA left over in the sterile soil matrix? Are there any results that can or should be reported there?

Author's response: "In our procedure the isolation of relic DNA from soil is minimized, because we first wash cells from the soil matrix and only then purify DNA from the collected cells (lines 408-419 in the Materials and Methods section). Of all of the soil microcosm samples that were sequenced, 70 {plus minus} 4% of the total read counts were attributed to a member of SynCom. However, this does not necessarily indicate that the remaining read counts were from "foreign" DNA, but potentially from sequences that were trimmed too short to match the search sequence or have small misreads in sequences that were not mapped to any SynCom member with the "grep" function."

Details of this should be included both in the Materials and methods and in the Results section. The other 30% of the total reads should also be taxonomically annotated to rule out contamination and/or artifacts from relic DNA

Reviewer #4 (Comments for the Author):

The authors have addressed all of the reviewers comments. However, there is some concern regarding the using absolute total abundance as calculated from 16S (accounting for live and dead cells) and total cfu/g (accounting for live cells) as this may skew the data. This reviewer recommends reporting 16S and total cfu/g separately or citing literature that validates the methodology.

Reviewer Response #2 for “Phosphite as an Engineered Niche for *Pseudomonas veronii* in a Synthetic Soil Bacterial Community

Reviewer comments (in italics) are shown below, followed by the author response and manuscript edits (in red).

There was also a comment about referencing methods/protocols by embedding a URL. Please be more robust in your citation here by either providing a scientific reference for the material and/or detailing methodological steps performed. The URL is not sufficient.

We apologize for the inappropriate notation, the following reference has been added under the “Community 16S rRNA Gene Amplicon Sequencing” section of the Materials and Methods.

Libraries were prepared according to the Illumina 16S Metagenomic Sequencing Library protocol (4) (https://support.illumina.com/content/dam/illumina-support/documents/documentation/chemistry_documentation/16s/16s-metagenomic-library-prep-guide-15044223-b.pdf) for the V3 to V4 region of the 16S rRNA gene (Primers provided in **Supplemental Table 2**).

[in References]

4. Illumina. 2013. *16S metagenomic sequencing library preparation: preparing 16S ribosomal RNA gene amplicons for the Illumina MiSeq system*. Illumina, San Diego, CA.

Reviewer 2

*As stated in my first review, the authors relied heavily on 16S rRNA amplicon sequencing to track SynCom composition and shifts in community structure, providing a detailed view of microbial dynamics. However, most of these amplicon sequencing results were presented in supplementary figures, making it difficult to fully interpret their significance alongside the main results presented in the manuscript. I suggested that authors should improve clarity and support the study's conclusions by moving some key supplementary figures into subpanels of the main figures; particularly those showing *P. veronii* ptxD⁺ relative abundance alongside SynCom members in the Grow and Stable conditions.*

This was not really addressed. I see that there is an updated Figure 3, but the main text frequently requires the reader to go into the supplementary information to validate main results of the paper. I highly suggest that the authors do a more careful screening on what data is needed to support the results and show this as a subpanel or full figure.

We thank the reviewer for this comment. We thought we had improved this by including the new Figure 3, but this was insufficient in their eyes. We now further included previous SI Figure 2 as new panel in Figure 2, and SI Figure 12 as new panel in Figure 6, which we hope improves readability. We have further considered which primary data would be needed to be shown in the main text, and which data provide further additional information and can be kept as SI Figures.

L94 the study's SynCom is reported to have "reproducible assembly and successional patterns". This should be cited or shown more rigorously in the Supplementary data. The level of experimental replication is not clear in Supplemental figures 2 & 3.

We apologize and this has been cited appropriately (Ref 23). Individual replicate variation is shown in e.g., Figure 3 panels, but also in all NMDS plots, which show (to our interpretation) clear replicate clustering.

I do not see where this citation specifically applies to the system studied here. I don't think its appropriate to simply generalize that soil community succession and assembly is deterministic and not context dependent. I suggest to either eliminate this claim or do analyses that would be included in the main text that support this as a result .

We thank the reviewer for the comment. We refer to this paper, because it shows that much of typical soil composition variation can be avoided by culturing soil microbiota under controlled soil microcosm conditions. This enables us to test more specifically the effects of nutrient availability and timing of inoculations. See, also, Causevic, S., et al. (2024). "Niche availability and competitive loss by facilitation control proliferation of bacterial strains intended for soil microbiome interventions." *Nat Commun* 15(1): 2557 (Ref. 9, mentioned in the context of strain inoculation in l. 343)

Details of this should be included both in the Materials and methods and in the Results section. The other 30% of the total reads should also be taxonomically annotated to rule out contamination and/or artifacts from relic DNA

We thank the reviewer for the comment, and hope the following explanation and additional analysis alleviates concern. Since we work with a defined SynCom where all members have been genome sequenced and assembled, we don't use OTU assignments, but directly 'grep' among the sequences for the known signatures of all members. This is then further corrected by the 16S rRNA gene copy numbers in the respective genomes (as explained in the Supplementary methods section "Community 16S rRNA Gene Amplicon..."). As a result, misreads that do not match the known sequence exactly are not assigned to a SynCom member, but do not originate from a contaminant.

To provide evidence of this, we examined a representative sample and timepoint: the SynCom diluted in fresh soil matrix (grow phase) inoculated with *P. veronii ptxD*⁺ and phosphite, sampled after 7 days (see Figure 3A). We extracted all the merged, paired-end reads that were unassigned by the 'grep' function, denoised and clustered with DADA2, and finally performed taxonomy assignment with QIIME2 (using SILVA database 13.8). The resulting assignments are shown in the table below. The majority of the assignments are to SynCom members, and the assignments that were not assigned to a SynCom member are highlighted in yellow. These all belong to the *Yersiniaceae* family and have notably low consensus scores. These hits could possibly originate from SynCom member *Rahnella* (the assignments with consensus scores 0.7-0.8 only identify down to the family level). Based on this in-depth example, we believe our methodology did not

introduce any meaningful contaminants or relic DNA, and respectfully suggest that no further changes are needed.

Kingdom	Phylum	Class	Order	Family	Genus	Species	Consensus
Bacteria	Pseudomona	Gammaprote	Pseudomona	Pseudomona	Pseudomona	NA	0.909
Bacteria	Pseudomona	Gammaprote	Pseudomona	Pseudomona	Pseudomona	NA	0.909
Bacteria	Actinomycet	Actinobacter	Micrococcal	Microbacteri	Microbacteri	NA	0.909
Bacteria	Pseudomona	Gammaprote	Enterobacter	Yersiniaceae	Ewingella	Ewingella am	0.545
Bacteria	Pseudomona	Alphaproteot	Hyphomicrot	Xanthobacte	Tardiphaga	NA	0.727
Bacteria	Pseudomona	Gammaprote	Lysobacteral	Lysobacterac	Lysobacter	NA	0.909
Bacteria	Pseudomona	Gammaprote	Pseudomona	Pseudomona	Pseudomona	NA	1
Bacteria	Pseudomona	Alphaproteot	Caulobacter	Caulobacter	Caulobacter	NA	0.909
Bacteria	Pseudomona	Gammaprote	Enterobacter	Yersiniaceae	NA	NA	0.7
Bacteria	Pseudomona	Gammaprote	Pseudomona	Pseudomona	Pseudomona	Pseudomona	0.7
Bacteria	Bacillota	Bacilli	Paenibacillal	Paenibacilla	Cohnella	NA	0.909
Bacteria	Pseudomona	Alphaproteot	Caulobacter	Caulobacter	Caulobacter	NA	0.909
Bacteria	Pseudomona	Gammaprote	Lysobacteral	Lysobacterac	Lysobacter	NA	0.909
Bacteria	Pseudomona	Gammaprote	Pseudomona	Pseudomona	Pseudomona	Pseudomona	0.636
Bacteria	Pseudomona	Gammaprote	Pseudomona	Pseudomona	Pseudomona	NA	0.909
Bacteria	Pseudomona	Gammaprote	Lysobacteral	Lysobacterac	Lysobacter	Lysobacter c.	0.636
Bacteria	Pseudomona	Gammaprote	Enterobacter	Yersiniaceae	Rahnella	NA	0.8
Bacteria	Pseudomona	Gammaprote	Pseudomona	Pseudomona	Pseudomona	NA	1
Bacteria	Pseudomona	Gammaprote	Pseudomona	Pseudomona	Pseudomona	NA	1
Bacteria	Pseudomona	Gammaprote	Pseudomona	Pseudomona	Pseudomona	NA	1
Bacteria	Pseudomona	Gammaprote	Pseudomona	Pseudomona	Pseudomona	Pseudomona	0.9
Bacteria	Pseudomona	Gammaprote	Enterobacter	Yersiniaceae	Serratia	NA	0.545
Bacteria	Pseudomona	Gammaprote	Burkholderia	Comamonad	Variovorax	NA	0.818
Bacteria	Pseudomona	Gammaprote	Enterobacter	Yersiniaceae	NA	NA	0.8
Bacteria	Pseudomona	Gammaprote	Lysobacteral	Lysobacterac	Rehaibacteri	NA	0.6
Bacteria	Pseudomona	Gammaprote	Pseudomona	Pseudomona	Pseudomona	NA	0.909
Bacteria	Pseudomona	Gammaprote	Pseudomona	Pseudomona	Pseudomona	Pseudomona	0.8
Bacteria	Pseudomona	Gammaprote	Lysobacteral	Lysobacterac	Lysobacter	NA	0.909
Bacteria	Pseudomona	Alphaproteot	Hyphomicrot	Devosiaceae	Devosia	NA	0.909
Bacteria	Pseudomona	Alphaproteot	Hyphomicrot	Xanthobacte	Tardiphaga	NA	0.7
Bacteria	Pseudomona	Alphaproteot	Caulobacter	Caulobacter	Caulobacter	NA	0.636
Bacteria	Pseudomona	Gammaprote	NA	NA	NA	NA	0.909

Reviewer 4

The authors have addressed all of the reviewers comments. However, there is some concern regarding the using absolute total abundance as calculated from 16S (accounting for live and dead cells) and total cfu/g (accounting for live cells) as this may skew the data. This reviewer recommends reporting 16S and total cfu/g separately or citing literature that validates the methodology.

We thank the reviewer for the comment. We detail this procedure in a new paragraph (1. 422-431). Importantly, since we analyze genomic signatures and viability in the same washed cell samples from the soil microcosms, and all SynCom members are culturable, we can multiply relative abundances by CFU/ml to have absolute population data.

Re: mSystems00061-25R2 (Phosphite as an Engineered Niche for *Pseudomonas veronii* in a Synthetic Soil Bacterial Community)

Dear Dr. Clara Bailey:

Your manuscript has been accepted, and I am forwarding it to the ASM production staff for publication. Your paper will first be checked to make sure all elements meet the technical requirements. ASM staff will contact you if anything needs to be revised before copyediting and production can begin. Otherwise, you will be notified when your proofs are ready to be viewed.

Sincerely,
Hans Bernstein
Editor
mSystems